# Sensitivity of climate effects of hydrogen to leakage size, location, and chemical background

Ragnhild Bieltvedt Skeie[1], Marit Sandstad[1], Srinath Krishnan[1], Gunnar Myhre[1], and Maria Sand[1]

[1]CICERO Center for International Climate Research, Oslo, Norway

*Correspondence to*: Ragnhild Bieltvedt Skeie (r.b.skeie@cicero.oslo.no)

**Abstract.** Use of hydrogen can reduce carbon dioxide emissions by replacing fossil fuel used as an energy carrier and reactant in metal production. When hydrogen is used, some hydrogen will leak during production, storage, transport, and end use. Via OH-induced reactions in the atmosphere, the hydrogen will enhance methane, ozone, and stratospheric water vapor in the atmosphere and hence increase the radiation imbalance. A recent multi-model study found the Global Warming Potential over a 100-year time horizon (GWP100) for hydrogen to be 11.6 ±2.8 (one standard deviation). Here, we use a chemistry transport model to investigate the sensitivity of GWP100 to the magnitude and the location of the hydrogen emission and the chemical composition of the background atmosphere. We show that the hydrogen GWP100 is independent of the size of the emission perturbation; is not dependent on where emissions occur except sites far from soil sink active areas; and is not very different for possible future chemical compositions of the atmosphere. For methane the $CH_4$ GWP100 increases by up to 3.4 for different future atmospheric compositions compared to present-day. Overall, the changes in the hydrogen GWP100 are within one standard deviation of the multi-model GWP100, except for emission perturbations at two distant sites not relevant for a future hydrogen economy. Therefore, when assessing emissions at different locations or for a future with different atmospheric composition than present-day, it is not necessary to adjust the multi-model GWP values.

## 1 Introduction

In a low-carbon economy, molecular hydrogen is expected to play a role as an energy carrier (IEA, 2023;HydrogenCouncil, 2023;DNV, 2022). Hydrogen is not a greenhouse gas and when it is used in a fuel cell to generate energy, only water vapor is emitted. However, during production, storage, transport and end-use some hydrogen may leak (Esquivel-Elizondo et al., 2023). The leaked hydrogen will react in the atmosphere with the hydroxyl radical (OH) and alter the abundance of other greenhouse gases: methane ($CH_4$), ozone ($O_3$), and water vapor in the stratosphere (strat. $H_2O$) (Prather, 2003). The climate benefits of replacing fossil fuel with hydrogen will depend on how much hydrogen is leaked, in addition to emissions related to the energy used to produce hydrogen (van Ruijven et al., 2011).

The climate effect of emissions of different gases can be compared using climate emission metrics (Forster et al., 2021). A commonly used metric is the Global Warming Potential (GWP). The GWP is defined as the ratio of the time integrated effective radiative forcing (ERF) over a given time horizon to a 1 kg pulse emission relative to that for 1 kg of $CO_2$. A time

horizon of 100 years is often used, GWP100. A shorter time horizon will give more weight to short-lived climate components (Fesenfeld et al., 2018;Shindell et al., 2017), such as $H_2$ with a lifetime of about 2 years (Ehhalt and Rohrer, 2009). A longer time horizon like 100 years, will give more weight to $CO_2$ and therefore be more appropriate for evaluating hydrogen as a replacement for $CO_2$ for reaching a long-term temperature stabilization target. The GWP100 is also used in an alternative method for evaluating the $CO_2$ mitigation potential GWP* (Cain et al., 2019;Allen et al., 2016) that better captures the temperature response of short- and long-lived greenhouse gases (Allen et al., 2018;Lynch et al., 2020).

A $H_2$ GWP100 of $11.6 \pm 2.8$ was found in the multi-model study by Sand et al. (2023); $12 \pm 6$ using UK Earth System Model (UKESM1) chemistry–climate model (Warwick et al., 2023) and $12.8 \pm 5.2$ in Hauglustaine et al. (2022) based on GFDL-AM4.1 model results (Paulot et al., 2021). Studies that only include changes in the troposphere find smaller GWP100 values (Derwent, 2023;Derwent et al., 2001;Field and Derwent, 2021;Derwent et al., 2020).

The largest source of uncertainty in the calculated GWP100 is the soil sink (Sand et al., 2023) and it is the largest - and most uncertain - term in the hydrogen budget (Ehhalt and Rohrer, 2009;Paulot et al., 2021). The soil sink varies geographically as it depends on the soil temperature and moisture as well as the biological activity (e.g. Paulot et al., 2021;Xiao et al., 2007). Due to greater landmass the soil sink is stronger in the Northern Hemisphere than in the Southern Hemisphere. This results in lower concentrations of hydrogen in the Northern Hemisphere, despite emissions dominating in the Northern Hemisphere. Using a two-dimensional tropospheric chemistry-transport model (TROPOS), Derwent (2023) studied the sensitivity of the $H_2$ GWP to the latitudinal dependence of the hydrogen emission pulse and found largest GWP values in the southernmost latitudes and weakest GWP values in the northernmost latitudes.

In addition to the soil sink, $H_2$ is removed from the atmosphere by chemical reaction with OH.

$$H_2 + OH \ \rightarrow \ H_2O + H \qquad\qquad (1)$$

Photochemical reaction with OH is the dominant sink of methane, so when OH reacts with hydrogen (Eq.1) this will reduce the available OH for methane loss and therefore enhance the atmospheric lifetime of methane. The OH concentration is dependent on the methane levels and other components such as carbon monoxide (CO) and non-methane volatile organic compounds (NMVOCs) that react with OH and hence reduce the OH levels. Nitrogen oxides ($NO_x$) on the other hand, lead to photochemical production of OH in the atmosphere. The $NO_x$ to CO emission ratio has been shown to be important for explaining the OH time evolution and changes in methane lifetime over time (Dalsøren et al., 2016;Skeie et al., 2023). The resulting H in Eq. 1 is involved in the complex photochemical ozone production. The ozone production is dependent on the concentrations of the ozone precursors methane, CO, NMVOCs and $NO_x$ in the atmosphere (Monks et al., 2015). Changes in atmospheric composition of these reactive gases influence the atmospheric lifetime of hydrogen and methane as well as the chemical production of ozone.

In this study we test the robustness of the GWP100 results presented in previous studies where hydrogen concentrations or hydrogen emissions are enhanced globally (Warwick et al., 2023;Hauglustaine et al., 2022;Sand et al., 2023). Here, we test the sensitivity to the geographical location of the hydrogen perturbation by adding 1 Tg yr$^{-1}$ hydrogen emissions to seven different point locations using a three-dimensional atmospheric chemistry transport model. We also test the linearity of the calculated GWP100 (i.e. that the obtained GWP100 is independent of the size of the hydrogen emissions) by changing the magnitude of the emission perturbation in the model simulations. Finally, the atmospheric composition of chemically active species might be different in the future, which will influence the chemical loss of hydrogen through changes in OH concentrations as well as the chemical production of hydrogen in the atmosphere. Therefore, we investigate the sensitivity of the calculated GWP100 on the chemical composition of the atmosphere using anthropogenic emissions and methane concentrations from three different Shared Socioeconomic Pathways (SSPs). In addition, based on the same model simulations, we calculate the $CH_4$ GWP100 and investigate how it differs for different chemical compositions of the atmosphere.

## 2 Methods

In this study we use an emission perturbation approach to calculate the GWP100 of hydrogen. We use the global chemical transport model OsloCTM3 to investigate the sensitivity of the calculated GWP100 due to the size and location of the hydrogen perturbation, as well as future atmospheric chemical composition.

### 2.1 GWP calculation

The $H_2$ GWP is the ratio of the absolute global warming potential (AGWP) for hydrogen relative to that for $CO_2$. The AGWP is defined as the time integrated effective radiative forcing of a 1 kg pulse emission over a given time horizon (Myhre et al., 2013). For a 100 year time horizon all the perturbations from an initial hydrogen pulse have decayed, and it is shown that a steady state perturbation matches the integrated response of a pulse emission (Prather, 2002;Prather, 2007).

To calculate the $H_2$ GWP100, a control simulation and a simulation with enhanced hydrogen emissions are run to steady state. From the perturbed hydrogen simulation, the change in atmospheric composition of $O_3$, $CH_4$, and strat. $H_2O$ and the resulting ERFs due to these changes are calculated. As emission driven simulations of methane are challenging due to large uncertainties in the methane sources and sinks (Saunois et al., 2020), global chemical models fix the surface concentration of methane. The change in the atmospheric methane is calculated from the modeled change in methane lifetime. As changes in methane also change the composition of $O_3$ and strat. $H_2O$ a methane perturbation experiment is also needed. From the methane perturbation experiment, where hydrogen is the same as in the control simulation, the atmospheric composition changes of $O_3$ and strat. $H_2O$, and hence the ERF, due to the changes in the methane lifetime in the hydrogen perturbation

experiment can be extracted. The contributions to the H$_2$ GWP100 from the changes in the methane lifetime are referred to as "methane induced". From the methane perturbation experiments performed, we also calculate the CH$_4$ GWP100.

We follow the same approach as in the multi-model study by Sand et al. (2023). While the hydrogen concentration was perturbed in the main simulations in that study, we use hydrogen emission perturbations and investigate the sensitivity of the GWP100 to how the perturbations are implemented in the model.

## 2.2 Sensitivity experiments

Three sets of sensitivity tests are performed. In the first set of sensitivity tests, we test if the calculated GWP100 is dependent on the size of the emission perturbation. The anthropogenic emissions of hydrogen are scaled so that the global emissions increase by 0.1, 1, 10 and 100 Tg yr$^{-1}$ (anthro01, anthro1, anthro10 and anthro100).

The second set of sensitivity tests investigate the dependence on the geographical location of the perturbations. Point source
emissions of 1 Tg yr$^{-1}$ are added to seven locations (Fig. 1) expected to span a wide range of possible model responses. As the soil sink is only active over land, we choose point emissions over the point furthest from land (nemo) located in the South Pacific; the most landlocked point (epia); a point in the US in an area with large soil sink velocities in the model (usdrydep); a point in East Africa close to the Equator where the soil sink velocity is large (lowlatdep); a more typical industrial point over Europe (munich); in addition to an Arctic (zep); and an Antarctic (maud) point (Table 1). The sites are
indicated by stars in Fig. 1.

For these first two sets of sensitivity tests, the same methane perturbation simulation enhancing 2010 methane surface concentration by 10%, was used to gauge the methane induced changes (Table 2). Note that the methane perturbation and the sensitivity tests for hydrogen perturbations correspond to two different control simulations, as hydrogen is concentration driven in the methane perturbation simulation and emission driven in the hydrogen perturbation simulations.

The last set of sensitivity tests investigate the sensitivity of the calculated GWP100 to the atmospheric composition. We use the same setup as for the other simulations but use methane surface concentration (Meinshausen et al., 2020) and gridded anthropogenic emissions for 2050 from three different SSPs (Gidden et al., 2019): SSP119, SSP434 and SSP585. The SSPs chosen were based on high and low methane concentration (Fig. 2a) and high and low NO$_x$ to CO emission ratio (Fig. 2b) as it is found to be a main driver of change in OH and methane lifetime (Dalsøren et al., 2016;Skeie et al., 2023) and would also
influence the atmospheric lifetime of hydrogen. For each SSP, a control simulation, a 10 Tg yr$^{-1}$ hydrogen emission increase simulation (as anthro10) (Table 1) and a 10% increase in surface methane concentration (Table 2) simulation are performed.

## 2.3 Model description

The OsloCTM3 (Søvde et al., 2012) is a chemistry transport model driven by 3-hourly meteorological forecast data generated by the Open Integrated Forecast System (OpenIFS, cycle 38 revision 1) at the European Centre for Medium-Range Weather Forecasts (ECMWF), and the horizontal resolution is ~2.25 × 2.25° with 60 vertical layers ranging from the surface and up to 0.1 hPa. The OsloCTM3 model is used in a similar set up as in Sand et al. (2023), except that the calculation of stratospheric water vapor has been updated to ensure a balanced hydrogen budget in the stratosphere and the geographical distribution of the anthropogenic emissions of hydrogen that was shifted 180 degrees has been corrected. The hydrogen soil sink scheme takes into account the soil moisture effect on dry-deposition velocities that depend on vegetation types (Sanderson et al., 2003) with no uptake for snow covered land and reduced uptake rate for cold surfaces (Price et al., 2007). For the sensitivity test with present-day atmospheric composition, the anthropogenic emissions used are from the Community Emissions Data System (CEDS) (version 2017-05-18) (Hoesly et al., 2018), biomass burning emissions from Global Fire Emissions Database (GFED4s, van der Werf et al., 2017) and the hydrogen emissions from Paulot et al. (2021). Emissions for the year 2010 were used from all three datasets. For the sensitivity tests with different atmospheric backgrounds, the anthropogenic emissions used are 2050 emissions from the three SSP scenarios (Gidden et al. (2019), SSP119: IAMC-IMAGE-ssp119-1-1, SSP434: IAMC-GCAM4-ssp434-1-1, SSP585: IAMC-REMIND-MAGPIE-ssp585-1-1), while hydrogen emissions and biomass burning emissions are kept the same as for the present-day simulations. The model is run with 2009 and 2010 meteorology repeatedly for in total 26 years, except for the methane concentration perturbations that were run for 20 years.

## 2.4 Forcing calculation

The forcing components included in the GWP100 calculations are $CH_4$, $O_3$ and strat. $H_2O$. The ERFs for these components are calculated from the last year of the OsloCTM3 steady-state simulations. The $CH_4$ ERF is calculated using a concentration-to-forcing factor of 0.448 mW m$^{-2}$ ppb$^{-1}$ (Etminan et al., 2016) with an adjustment term of −14% (Forster et al., 2021), where the adjustment term converts the forcing from stratospheric temperature adjusted radiative forcing to ERF. The ERF of $O_3$ is calculated using monthly mean three-dimensional $O_3$ fields in the perturbations compared to the control simulation multiplied by a monthly three-dimensional kernel for $O_3$ radiative forcing (RF) that includes stratospheric temperature adjustments (Skeie et al., 2020). No tropospheric adjustments are assumed, and the resulting radiative forcing values are treated as ERF. For strat. $H_2O$, the RF values calculated offline using radiative transfer schemes for longwave and shortwave radiation separately (Myhre et al., 2007) are also treated as ERF, assuming no tropospheric adjustments.

**Table 1: List of sensitivity tests for the hydrogen emission perturbation. In the sensitivity tests, the anthropogenic hydrogen emissions (anthro) are perturbed. Each of the SSP-sensitivity tests have separate corresponding control simulations, while the tests with different magnitude of the emission perturbations and the tests with different geographical point emission perturbations share the same control simulation.**

| Name of sensitivity test | Description of test |
| --- | --- |
| anthro01 | anthro+0.1 Tg yr$^{-1}$ |
| anthro1 | anthro+1 Tg yr$^{-1}$ |
| anthro10 | anthro+10 Tg yr$^{-1}$ |
| anthro100 | anthro+100 Tg yr$^{-1}$ |
| epia | anthro+1Tg yr$^{-1}$ in the middle of the continents (46.17 N, 85.58 E) |
| maud | anthro+1Tg yr$^{-1}$ in Antarctica (72.3 S, 12 E) |
| lowlatdep | anthro+1Tg yr$^{-1}$ where model soil sink is largest (3.3 N, 41.0E) |
| munich | anthro+1Tg yr$^{-1}$ in central Europe (48.1 N, 11.6 E) |
| nemo | anthro+1Tg yr$^{-1}$ in the middle of the ocean (48.5 S, 123.2 W) |
| usdrydep | anthro+1Tg yr$^{-1}$ in US where soil sink is large (34.8 N,100.7 W) |
| zep | anthro+1Tg yr$^{-1}$ in the Arctic (78.5 N, 11.56 E) |
| SSP119 | anthro+10 Tg yr$^{-1}$ in 2050 SSP1-1.9 chemical atmosphere |
| SSP434 | anthro+10 Tg yr$^{-1}$ in 2050 SSP4-3.4 chemical atmosphere |
| SSP585 | anthro+10 Tg yr$^{-1}$ in 2050 SSP5-8.5 chemical atmosphere |

**Table 2: List of methane sensitivity tests. The Present-day sensitivity test is used for all size and location perturbations, while each SSP test has a corresponding SSP methane sensitivity test.**

| Name of sensitivity test | Description of test |
| --- | --- |
| Present-day | 10% increase in surface concentration from 1813 ppbv in the control |
| SSP119 | 10% increase in surface concentration from 1427 ppbv in the control |
| SSP434 | 10% increase in surface concentration from 2223 ppbv in the control |
| SSP585 | 10% increase in surface concentration from 2446 ppbv in the control |

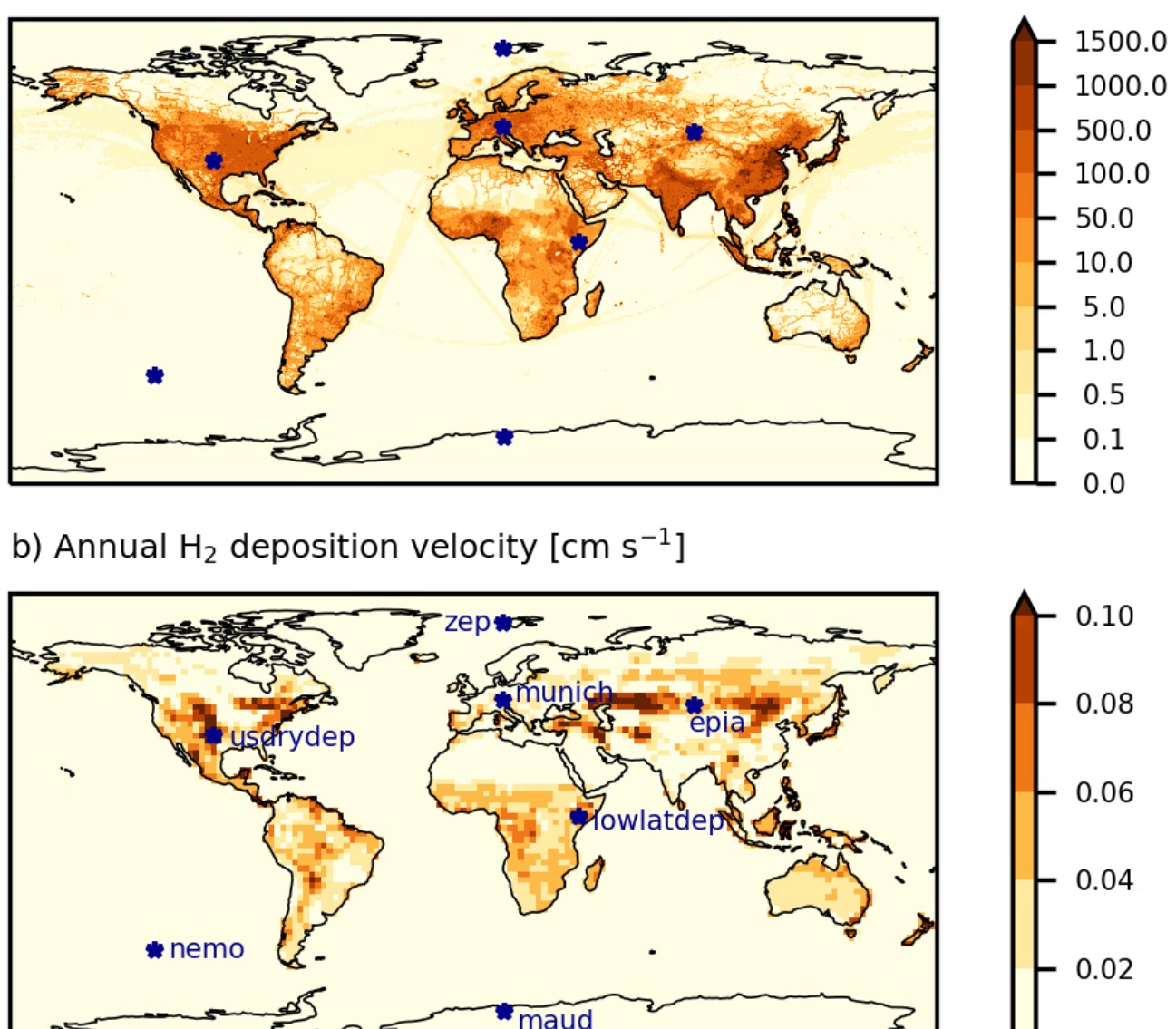

**Figure 1:** In a) the total annual anthropogenic emissions of hydrogen from Paulot et al. (2021) for 2010 is shown and in b) the annual mean deposition velocity of hydrogen from OsloCTM3 present-day control simulation is shown. In both figures, the sites where 1 Tg yr$^{-1}$ is added in the sensitivity tests for perturbation locations are indicated by a blue star and in b) the names of the respective sensitivity tests are also indicated.

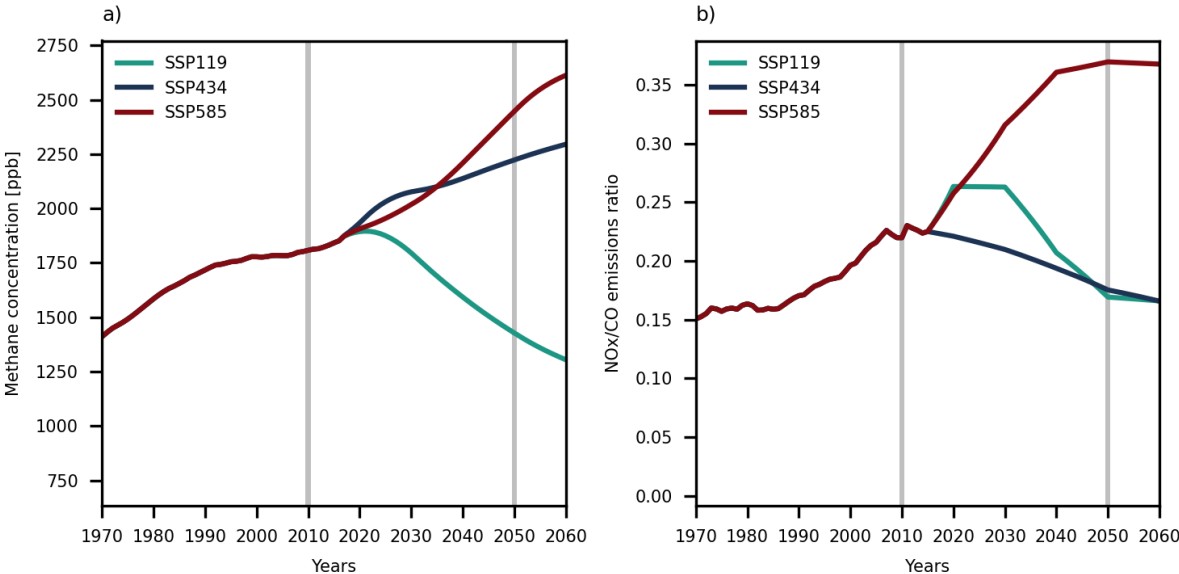

**Figure 2: In a) the historical and future methane surface concentrations and in b) the historical and future anthropogenic NOₓ to CO emission ratios. The three SSPs used in this study are shown. The years used in the simulations (2010 and 2050) are indicated by a vertical line. The anthropogenic emissions for NOₓ and CO separately, in addition to volatile organic components (VOCs) are shown in Fig. S1.**

## 3 Results

The resulting $H_2$ GWP100 values from all the sensitivity tests are shown in Fig. 3. The individual contributions from methane are in green; ozone in yellow; and stratospheric water vapor in purple. The hashed areas show how much of the change is due to changes in methane lifetime. Also added to the figure is the multi-model $H_2$ GWP100 from Sand et al. (2023) where the uncertainty bar indicates one standard deviation uncertainty range based on spread in underlying values found in literature and from the model ensemble.

### 3.1 Is the response to emissions size linear?

The first set of sensitivity tests investigate the linearity in the chemical response with respect to the magnitude of the hydrogen emission perturbation. Adding a hydrogen perturbation of 0.1, 1, 10 or 100 Tg yr⁻¹ resulted in very similar GWP100 values ranging from 12.2 to 12.6 as seen in the first 4 bars in Fig. 3. To conclude, the GWP100 values are independent of the magnitude of the emission perturbation within the range 0.1 to 100 Tg yr⁻¹

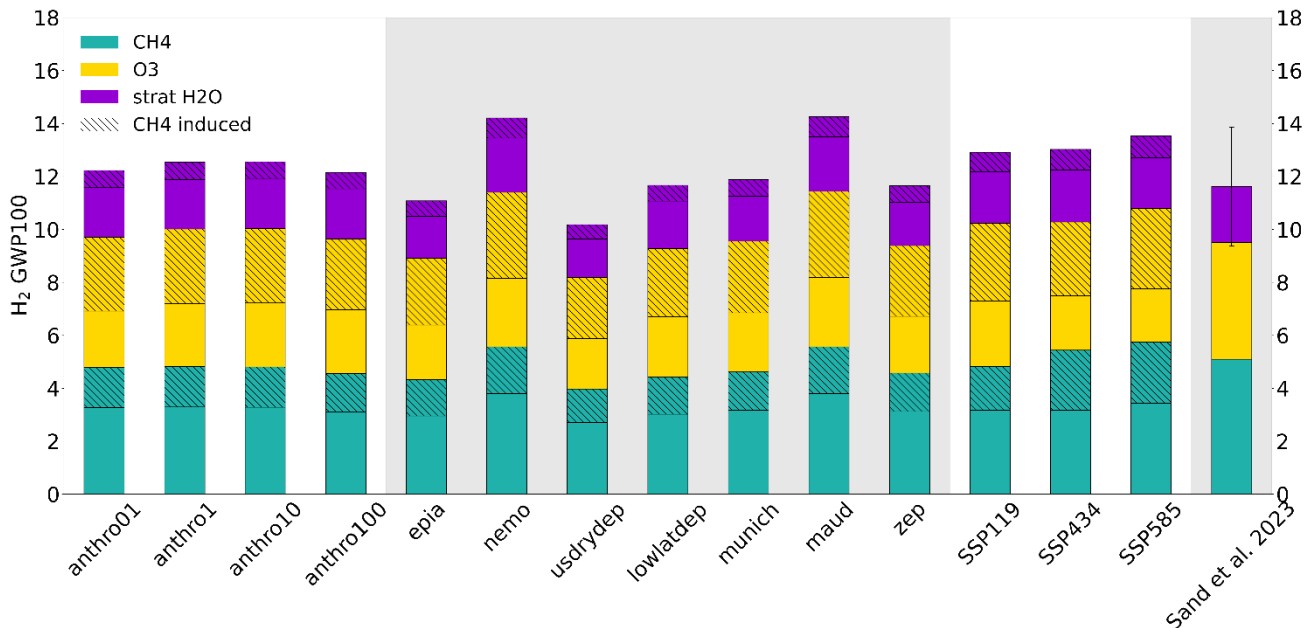

**Figure 3: The H₂ GWP100 for the different sensitivity tests (Table 1) where the individual contributions from methane (green), ozone (yellow), and stratospheric water vapor (purple) as well as methane induced changes in these (hashed) are shown. The multi model mean with uncertainty range (one standard deviation) assessed in Sand et al. (2023) is shown to the right. The GWP100 values are presented in Table S1.**

## 3.2 Does location matter?

The second set of sensitivity tests investigate the GWP100 sensitivity to the geographical location of the hydrogen emission perturbation. In anthro1, 1 Tg yr$^{-1}$ was added in the simulation by scaling the anthropogenic emissions while in these simulations, 1 Tg yr$^{-1}$ is added in seven specific point locations around the globe (Table 1). The point locations are carefully selected to span a wide range of possible GWP values based on distance to soil sink active areas and latitude (Fig. 1b). The 190 resulting GWP100 values range from 10.2 to 14.2 (Table S1), which is 19% lower and 14% higher than the anthro1 GWP100 of 12.5 (Fig. S2).

Figure 4a shows the increase in global mean surface hydrogen concentration per hydrogen flux for all sensitivity simulations. The hydrogen surface concentrations are highly dependent on where the hydrogen perturbation is added in the simulation (bars on grey background in Fig. 4a). Adding 1 Tg H₂ yr$^{-1}$ at the ocean site (nemo) and in Antarctica (maud) results in larger 195 increases in hydrogen concentrations of 7.9 and 9.2 ppb than adding the 1 Tg H₂ yr$^{-1}$ to areas with large soil sink in the model such as usdrydep and lowlatdep, which show an increase of 5.2 and 5.4 ppb respectively. The more typical industrial area

(munich) has a very similar change in atmospheric hydrogen per flux of hydrogen compared to the anthro1 simulation, both with 6.3 ppb per 1 Tg $H_2$ yr$^{-1}$.

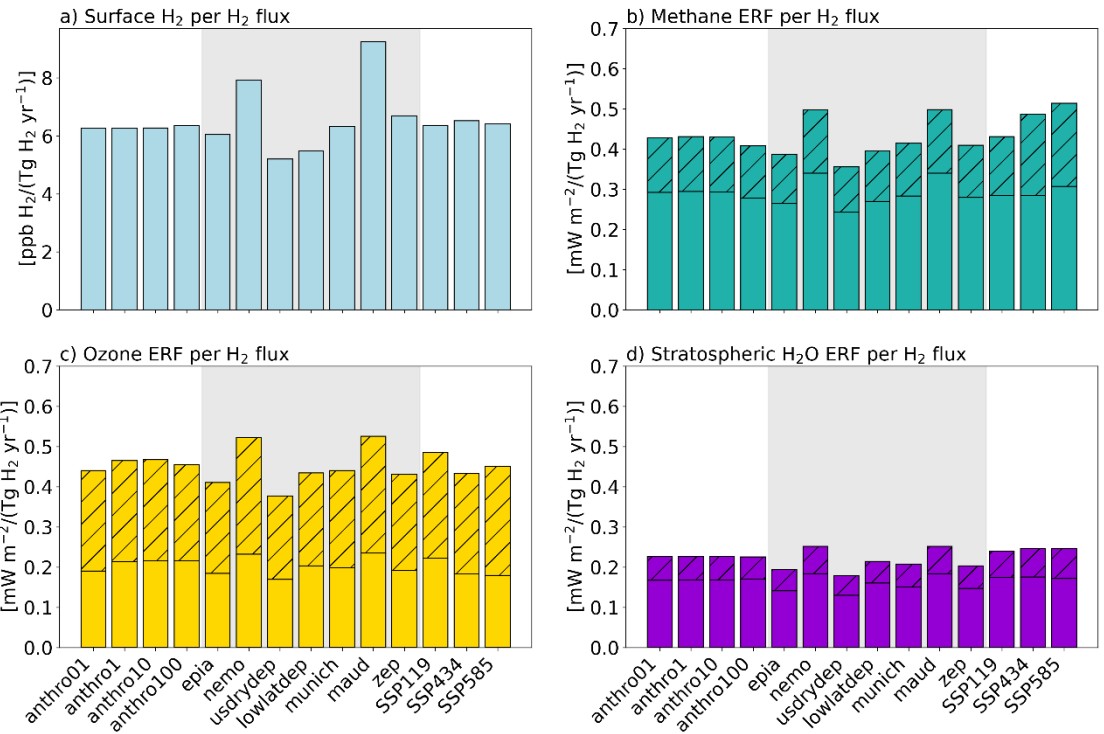

**Figure 4: Changes in a) surface hydrogen concentration, b) ERF of methane, c) ERF of ozone and d) ERF of stratospheric water vapor due to 1 Tg yr$^{-1}$ flux of hydrogen. The methane induced changes are indicated by the hatching. The underlying numbers for these figures are presented in Tables S2-S4.**

The difference in the increase of atmospheric hydrogen concentration per flux of hydrogen can be explained by different perturbation lifetimes. For chemically reactive species, such as methane, the chemical loss in the atmosphere via OH will be less efficient when more methane is added to the atmosphere, and the lifetime of the methane perturbation is enhanced. For methane the chemical feedback factor (the lifetime of the perturbation divided by the background lifetime of the atmospheric component) is larger than 1 (Holmes et al., 2013;Sand et al., 2023;Thornhill et al., 2020). For a hydrogen perturbation, the geographical distribution of the emission perturbation influences the perturbation lifetime (Table 3). For the atmospheric lifetime, the perturbation lifetime is larger than the lifetime, with the largest increase of 10% at the Antarctic site (maud). For the soil sink lifetime, the perturbation lifetime is shorter than the hydrogen background lifetime except for the ocean site (nemo) and Antarctic site (maud) (Table 3). The soil sink is enhanced when emissions are close to the soil sink active areas relative to the total soil sink and hence the perturbation lifetime is reduced. As the soil sink lifetime (3.5 years) is lower than the atmospheric lifetime (7.0 years) a similar change is seen for the total perturbation lifetime relative to total background

lifetime as for the soil sink lifetime (Table 3). The largest difference between the perturbation lifetime and the hydrogen
lifetime is for usdrydep with a difference of 0.57 years.

**Table 3: Total lifetime of hydrogen (total burden divided by the total loss in the control simulation), atmospheric lifetime (total burden divided by the atmospheric loss in the control simulation) and soil sink lifetime (total burden divided by the soil sink loss) and the respective perturbation lifetimes (change in burden divided by change in loss in the perturbation simulation relative to the**
**control simulation) for the different sensitivity tests. The change in perturbation lifetime relative to the lifetime in the control simulation (in %) are given in the parentheses.**

|  | $H_2$ total lifetime [yrs] | $H_2$ perturbation lifetime [yrs] | $H_2$ atmospheric lifetime [yrs] | $H_2$ atmospheric perturbation lifetime [yrs] | $H_2$ soil sink lifetime [yrs] | $H_2$ soil sink perturbation lifetime [yrs] |
|---|---|---|---|---|---|---|
| anthro01 | 2.35 | 2.17 (-7.5) | 7.02 | 7.23 (3.1) | 3.53 | 3.10 (-12) |
| anthro1 | 2.35 | 2.17 (-7.5) | 7.02 | 7.24 (3.2) | 3.53 | 3.10 (-12) |
| anthro10 | 2.35 | 2.17 (-7.4) | 7.02 | 7.27 (3.6) | 3.53 | 3.10 (-12) |
| anthro100 | 2.35 | 2.20 (-6.3) | 7.02 | 7.55 (7.6) | 3.53 | 3.10 (-12) |
| nemo | 2.35 | 2.56 (9.0) | 7.02 | 7.64 (8.9) | 3.53 | 3.85 (9) |
| epia | 2.35 | 1.94 (-17.2) | 7.02 | 7.25 (3.3) | 3.53 | 2.66 (-25) |
| munich | 2.35 | 2.08 (-11.3) | 7.02 | 7.24 (3.2) | 3.53 | 2.92 (-17) |
| usdrydep | 2.35 | 1.77 (-24.5) | 7.02 | 7.18 (2.3) | 3.53 | 2.35 (-33) |
| maud | 2.35 | 2.59 (10.2) | 7.02 | 7.73 (10.2) | 3.53 | 3.89 (10) |
| zep | 2.35 | 2.09 (-11.0) | 7.02 | 7.40 (5.5) | 3.53 | 2.91 (-17) |
| lowlatdep | 2.35 | 1.99 (-15.0) | 7.02 | 7.22 (2.9) | 3.53 | 2.75 (-22) |
| SSP119 | 2.37 | 2.20 (-6.9) | 7.26 | 7.59 (4.6) | 3.51 | 3.11 (-12) |
| SSP434 | 2.45 | 2.26 (-7.8) | 7.95 | 8.26 (3.9) | 3.54 | 3.11 (-12) |
| SSP585 | 2.41 | 2.22 (-8.0) | 7.55 | 7.80 (3.3) | 3.55 | 3.11 (-13) |


The longer the lifetime of the hydrogen perturbation, the larger the change in hydrogen burden in the atmosphere. A larger change in hydrogen burden leads to a larger forcing and forcing per flux of hydrogen (as shown in Fig. 4b for $CH_4$, Fig. 4c for $O_3$, and Fig. 4d for strat. $H_2O$) and larger GWP100. Similarly, a smaller change in atmospheric hydrogen per hydrogen
flux leads to smaller forcing and smaller GWP100 values. Although there is a spread in the GWP values of 4.1, the results are within the one sigma uncertainty range in the multi-model study by Sand et al. (2023) of 9.4 to 13.9, with the exception

of nemo and maud that are just outside this range, both with a GWP100 of 14.2. One should note that these two sites have an extreme location relative to what can be expected in a future hydrogen economy. For the continental sites more relevant for a future hydrogen economy (usdrydep, munich, lowlatdep, epia) the GWP100 range from 10.2 to 11.9, slightly less than

perturbing the total anthropogenic emissions (anthro1) of 12.5.

### 3.3 Does the chemical background matter?

In the future, the chemical composition of the atmosphere may be different than for present-day. As future emissions of chemically active components depend on technological developments and future societal evolution, different scenarios span possible pathways for emissions and concentrations of these reactive species.

Before looking at the sensitivity of the calculated GWP to the atmospheric chemical background, we will look at the hydrogen and methane budget in 2050. For all the SSP simulations we have used the same hydrogen emissions as in the present-day simulations (CNTR), so the changes in the hydrogen burden are only due to changes in atmospheric production and loss of hydrogen. Table 4 shows how the hydrogen budget changes in the three different SSP steady state simulations for 2050, relative to the 2010 steady state conditions used in the present-day simulation.

The hydrogen burden (as well as surface concentration) decreases in SSP119 mainly due to a decrease in the atmospheric production of hydrogen (Table 4). As chemical degradation of methane in the atmosphere is a main route of hydrogen production (Ehhalt and Rohrer, 2009;Paulot et al., 2021) the reduction in methane in SSP119 (Table 2, Fig. 2) as well as anthropogenic emissions of non-methane VOCs (Fig. S1), lower the amount of hydrogen in the atmosphere. In the two other scenarios, the methane levels are higher (Table 2, Fig. 2) and hydrogen production and burden increases (Table 4). Note that

we only change the atmospheric composition and do not change the climatic conditions in these simulations. As expected, the soil sink lifetime is similar in these simulations (Table 4) as the meteorology is the same.

**Table 4:** The hydrogen budget terms for present day (CNTR) and the change in these budget terms for the different SSPs relative to CNTR. The budget terms included are burden, surface concentration, atmospheric productions and the sink

represented as lifetimes. The hydrogen emissions are set equal in all the simulations.

| | $H_2$ burden [Tg] | $H_2$ surface concentration [ppbv] | $H_2$ atmospheric production [Tg yr$^{-1}$] | $H_2$ atmospheric lifetime [yrs] | $H_2$ soil sink lifetime [yrs] | $H_2$ total lifetime [yrs] |
|---|---|---|---|---|---|---|
| CNTR | 205 | 559 | 55.8 | 7.02 | 3.53 | 2.35 |
| SSP119 – CNTR | -23.6 | -63.9 | -10.8 | 0.24 | -0.01 | 0.02 |
| SSP434 – CNTR | 20.4 | 54.1 | 4.62 | 0.93 | 0.02 | 0.10 |

| | | | | | | |
|---|---|---|---|---|---|---|
| SSP585 – CNTR | 29.5 | 78.9 | 9.85 | 0.53 | 0.02 | 0.07 |

The atmospheric lifetime of hydrogen, calculated as the atmospheric burden divided by the chemical loss through OH (Eq. 1), depends on the chemical composition of the atmosphere. A larger concentration of methane will reduce the available OH, and the atmospheric lifetime of both methane and hydrogen will increase. In addition to methane, the $NO_x$ to CO emission ratio is also an important factor controlling OH in the atmosphere (Dalsøren et al., 2016). In all the scenarios the atmospheric lifetime of hydrogen increases (Table 4). For SSP434, both the increased methane concentration and the lower $NO_x$ to CO emission ratio (Fig. 2) push in the direction of less OH and a longer hydrogen atmospheric lifetime. This is the scenario where the hydrogen atmospheric lifetime increased the most by 0.9 years. In SSP119, the methane concentration is lower than for present day, contributing to a shorter hydrogen atmospheric lifetime, while the $NO_x$ to CO emission ratio (Fig. 2) pushes in the direction of a longer atmospheric lifetime. The resulting change in the hydrogen atmospheric lifetime is an increase by 0.2 years in SSP119 in 2050 compared to the present-day simulations. For SSP585, the methane and $NO_x$ to CO emission ratios change relative to present day also act in different directions with respect to OH and atmospheric lifetime of hydrogen. The hydrogen atmospheric lifetime increases by 0.5 years in this scenario in 2050 relative to present day. There are no scenarios with reduced methane concentration and higher $NO_x$ to CO emission ratios where both would have pushed in the direction of a shorter hydrogen atmospheric lifetime. As soil sink is the dominant loss term for hydrogen, the total lifetime change is smaller than the change in atmospheric lifetime, and the largest increase in total lifetime is for SSP434 with 0.1 years.

Table 5 shows the methane budget terms for present-day and the respective changes in the three SSPs. The change in the methane lifetime due to OH is similar to the change in the hydrogen atmospheric lifetime. As the total methane lifetime is dominated by the OH loss, the total lifetime of methane increases by 0.9 years in SSP434 relative to present day.

**Table 5: The methane budget terms for present-day (CNTR) and the change in these budget terms for the different SSPs relative to CNTR. For the total lifetime a soil sink lifetime of 160 years and a stratospheric lifetime (excluding OH loss) of 240 years is assumed as in Sand et al. (2023).**

| | $CH_4$ burden [Tg] | $CH_4$ surface concentration [ppbv] | $CH_4$ lifetime due to OH (whole atmosphere) [yrs] | Total $CH_4$ lifetime [yrs] | Feedback factor | Perturbation lifetime |
|---|---|---|---|---|---|---|
| CNTR | 4975 | 1813 | 7.38 | 6.85 | 1.46 | 10.04 |
| SSP119 – CNTR | -1059 | -386 | 0.29 | 0.25 | 0.05 | 0.72 |
| SSP434 – CNTR | 1129 | 410 | 1.05 | 0.90 | 0.25 | 3.24 |
| SSP585 – CNTR | 1738 | 633 | 0.61 | 0.52 | 0.21 | 2.31 |

The $H_2$ GWP100 for the different atmospheric composition sensitivity tests are shown in Fig. 3, ranging from 12.9 to 13.5 compared to 12.6 (anthro10) for 2010 atmospheric composition. For the geographical location sensitivity test (section 3.2), the $H_2$ GWP100 differed due to differences in the perturbation lifetimes. For the sensitivity tests with different chemical backgrounds, the perturbation lifetimes are similar to present day conditions (anthro10) (Table 3) and the changes in surface concentration of hydrogen per hydrogen flux are similar to anthro10 (Fig. 3a).

For the direct contribution to methane ERF per hydrogen flux (the area not hatched in Fig. 4b) the contribution is very similar for the different SSPs. For the methane induced changes (hatched part of the bars), there are larger differences, with larger contribution in SSP585 and SSP434 compared to present day (anthro10) and SSP119. The methane induced relative contribution to the GWP100 is hence larger in SSP585 and SSP434 compared to SSP119 (Table S5). The methane induced changes for methane ERF per hydrogen flux is estimated based on the methane feedback factor. The methane feedback

factor increases by 0.25 and 0.21 in SSP434 and SSP585 respectively, while in SSP119 the feedback factor only increases by 0.05 compared to present-day (Table 5).

    For ozone ERF per hydrogen flux, the direct (unhatched part in Fig. 4c) and the total (entire bar in Fig. 4c) contribution from hydrogen perturbation is larger in SSP119 compared to the two other SSPs. Ozone contributes 42% of the total GWP100 in SSP119 while in the two other scenarios it contributes 37% (Table S6). The changes in methane ERF and ozone ERF per

flux are compensating, which result in very similar GWP100 values for the different scenarios (Fig. 3) ranging from 12.9 in SSP119 to 13.5 in SSP585. The GWP100 increases by 3.0% to 7.8% when chemical background from three different SSPs for the year 2050 is used compared to using a present-day atmospheric composition in the simulations (Fig. S2).

    Based on the methane perturbation experiments, the GWP100 of methane can also be calculated, as was done in the multi-model study by Sand et al. (2023). The methane chemical feedback factor increased from a present-day value of 1.46 to 1.71

in SSP434, which is a main reason why the methane GWP100 is largest in this scenario (Fig. 5). The ozone contribution to the methane GWP100 is largest in SSP119, while the methane contribution to the GWP100 is smaller in SSP119 compared to the two other SSPs. Calculated $CH_4$ GWP100 increases by 6 to 13% in the SSPs for year 2050 relative to the present-day atmospheric composition. The GWP100 values of 28.6 to 30.5 and 27.0 for present day conditions (Table S8, Fig. 5) are however well within the uncertainty range of 19 to 35 from IPCC AR6 if the uncertainty in the AGWP for $CO_2$ is excluded

(Forster et al., 2021;Smith et al., 2021).

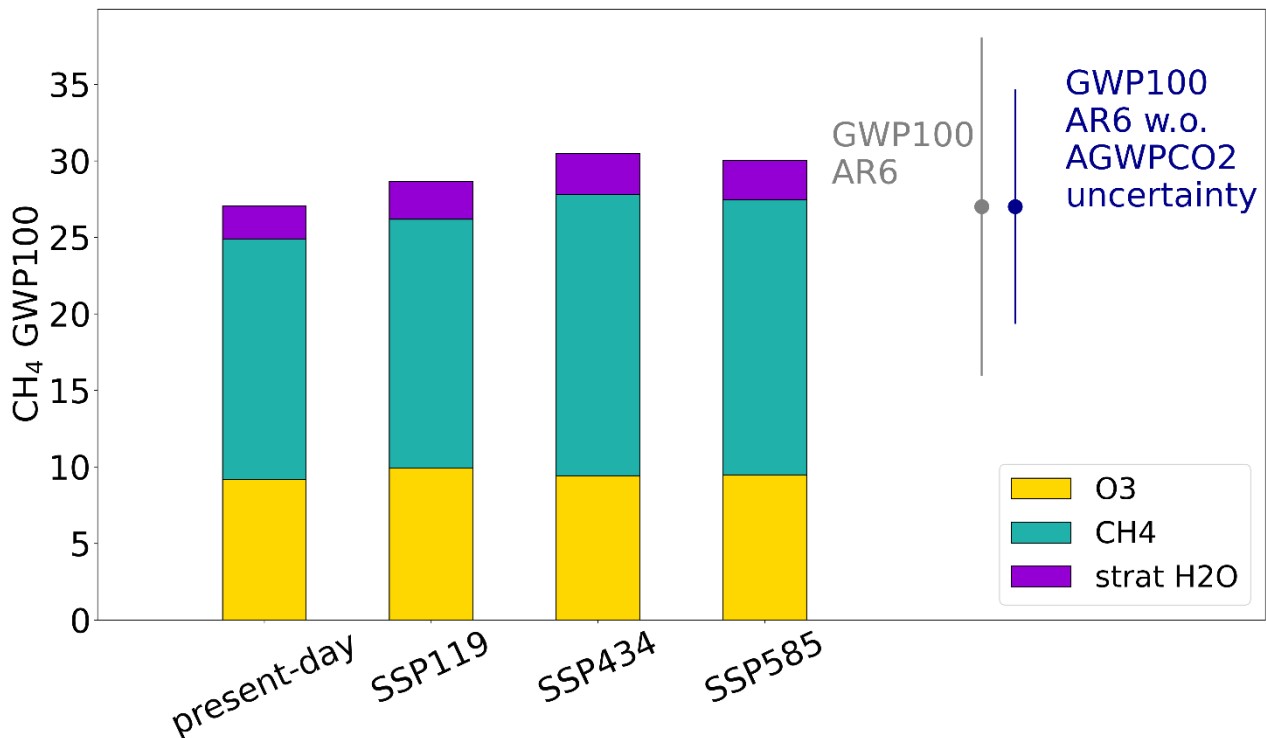

**Figure 5: GWP100 for methane calculated based on simulations with present-day and three SSPs atmospheric composition for 2050 with individual contributions from methane (green), ozone (yellow), and stratospheric water vapor (purple). The IPCC AR6 uncertainty range is the 5-95% range for GWP100 non-fossil fuel methane (Forster et al., 2021) and the range excluding uncertainties in AGWP for $CO_2$ (Smith et al., 2021). The GWP100 values are presented in Table S7.**

## 4. Discussion

In this study, we have used a chemical transport model to investigate the sensitivity of the $H_2$ GWP100 on how the hydrogen emission perturbation is added in the model simulations.

We find that GWP100 is independent of the size of the emission perturbation in the range of 0.1 to 100 Tg yr$^{-1}$ explored in this sensitivity study. A linear response with respect to the size of the perturbation was also found by Derwent (2023) using a two-dimensional tropospheric chemistry-transport model. Derwent (2023) investigated also the latitudinal dependence of the emission pulse on the calculated GWP. In the two-dimensional model, without longitudinal dimension, Derwent (2023) found that the GWP depended on latitude, with the largest GWP values in the southernmost latitude band and the smallest GWP values in the northernmost latitude band. This is also what we find for the Arctic (zep) and Antarctic (maud) emission locations with $H_2$ GWP100 of 11.7 and 14.2 respectively. The reason for the north-south gradient in the GWP is that the magnitude of the sink differs between the hemisphere due to different distribution of land and ocean areas (Derwent, 2023).

However, with a three-dimensional model, we do find dependencies on the longitudinal distribution as well and we find the distance to soil-sink active areas to be important for the calculated GWP.

The reason for the different GWP values in the geographical sensitivity tests can be explained by differences in the perturbation lifetime. For the sites nemo and maud, the perturbation lifetime is enhanced relative to the lifetime of hydrogen in the control simulation. For sites close to areas where hydrogen can be taken up by the soil, the perturbation lifetime is shorter than the hydrogen lifetime. This is because the soil sink is enhanced when emissions are close to the soil sink active areas relative to the total soil sink. In Sand et al. (2023) the perturbation lifetime relative to the background lifetime is

slightly less than one, and ranges from 0.95 to 1.0. These simulations were mostly concentration driven and hydrogen concentration was enhanced by 10% globally. For the geographical sensitivity tests, the range in perturbation lifetime was from 1.8 to 2.6 years, 25% shorter to 10% longer than the background lifetime. The GWP100 (10.2 to 14.2) values span the one standard deviation range from Sand et al. (2023), but note that some of the sites chosen are remote locations (in the Arctic, Antarctic, Southern Ocean) that are not relevant locations for future hydrogen economy. For continental sites more

relevant for a future hydrogen economy, the GWP100 values are well within the one standard deviation range assessed in Sand et al. (2023).

The hydrogen economy is expected to grow, and in the future, the atmospheric composition might be different than that of the 2010 conditions used to calculate GWP100. Therefore, we also investigated the sensitivity to the GWP100 by doing the emission perturbations on top of three different 2050 atmospheres based on SSP scenarios. The three different SSPs have

different combinations of $NO_x$ to CO emission ratios and methane levels that both influence the atmospheric lifetime of hydrogen. The atmospheric lifetime of both hydrogen and methane increased in all the scenarios, and in SSP434 by as much as ~1 year. Also, the methane chemical feedback factor increased from 1.46 to 1.71 (17%) in SSP434. A feedback factor of 1.7 is larger than what is found for present day atmospheres where the methane feedback factor in multi-model studies ranges from 1.36 to 1.55 (Sand et al., 2023), 1.34 ± 0.06 (Holmes et al., 2013) and 1.30 ± 0.07 (Thornhill et al., 2020).

Voulgarakis et al. (2013) also found an increase in the feedback factor from 1.24 for present day conditions to 1.50 in 2100 for a high methane low $NO_x$ scenario that included changes in climatic factors in addition to the changes in atmospheric chemistry composition. A larger feedback factor will enhance the methane induced contribution to the GWP100 (hatched green part in Fig. 3). As methane lifetime is low in this model (Table 5) compared to assessed estimates 9.1 ± 0.9 years (Szopa et al., 2021), the resulting methane feedback factor and chemical response following a perturbation, may be different

in a model with lower OH levels and longer methane lifetime.

From the methane perturbation experiments, the $CH_4$ GWP100 can also be calculated. The $CH_4$ GWP100 increased by 1.6 (6%) to 3.4 (13%) depending on the scenarios. These increases are driven by the increases in the perturbation lifetime of methane due to the change in the atmospheric composition of chemically reactive components in the scenarios. Note that changes in lightning emissions of $NO_x$, humidity, temperature and UV-radiation also alter the atmospheric oxidation capacity

(Nicely et al., 2018;Dalsøren et al., 2016;Voulgarakis et al., 2013) but future changes in these are not included here. Liu et al. (2024) investigated the future changes in the oxidation capacity in three global climate model simulations using two of the other SSPs. For a high mitigation scenario, with limited global warming, and reduced methane concentration (SSP126) the methane lifetime decreased by a range of 0.19-1.1 years due to increased OH concentration. For a low mitigation scenario with increased methane concentration and large temperature change (SSP370) the methane lifetime increased due to changes in the atmospheric composition partly masked by changes in climate, resulting in a total increase of 0.43 to 1.7 years from 2015 to 2100. Note that the $NO_x$ to CO anthropogenic emission ratios for 2050 are similar to the present-day ratio in these two scenarios. As noted above, the modelled methane lifetime is short and hence the OH level is high. Lower OH levels will enhance the lifetime of methane and the $CH_4$ GWP values but also impact the atmospheric chemistry involving ozone production. Further studies should investigate the role of OH levels on both $H_2$ and $CH_4$ GWP100 estimates.

For the $H_2$ GWP100, where the soil sink is the dominant control on the total lifetime, we find that the GWP100 values increase slightly compared to the present-day values when atmospheric composition is changed. The soil sink uncertainty is the main contributor to the uncertainty in $H_2$ GWP100 (Sand et al., 2023). There are large knowledge gaps in our understanding and representation of soil sink in models (Paulot et al., 2021) and hence knowledge gaps in understanding historical, present-day and future changes in the largest term in the hydrogen budget (Ehhalt and Rohrer, 2009). There is indication of an increase in soil sink over the recent years in the Northern Hemisphere due to changes in soil moisture, soil temperature and snow cover (Paulot et al., 2024). Future work should investigate potential changes in soil sink driven by climatic changes and assess how it will influence $H_2$ GWP100 values.

## 5. Conclusions

In this study we have used a chemistry transport model to investigate the sensitivity of the calculated $H_2$ GWP100 to the magnitude of the hydrogen emission perturbation, the location of the hydrogen emission perturbation as well as the chemical composition of the background atmosphere.

The $H_2$ GWP100 values are not dependent on the magnitude of the emission perturbation with values ranging from 12.2 to 12.6 but are somewhat dependent on the location of the hydrogen perturbation in the model with values ranging from 10.2 to 14.2. The further away from soil sink active areas, the larger the GWP100 values. The values fall within the one standard deviation uncertainty range estimated in Sand et al. (2023) with the exception of the two most extreme locations, in the southern Pacific and in Antarctica, where the values are slightly larger. One should note that these two locations are not relevant sites for hydrogen usage in a future hydrogen economy. For other short-lived components, emission location matters for the radiative response (Berntsen et al., 2006) and GWP values are reported regionally (Myhre et al., 2013;Aamaas et al., 2016). The results here indicate that this is not necessary for hydrogen.

The methane levels and $NO_x$ to CO emission ratio influence the oxidation capacity of the atmosphere and hence the lifetime of both methane and hydrogen. We have investigated how the $H_2$ GWP100 depends on the chemical composition of the atmosphere using SSP scenarios. In SSP585, where methane levels in 2050 are 35% above 2010 levels, the $H_2$ GWP100 is 13.5 and close to the upper range of the uncertainty range (+1 standard deviation) in Sand et al. (2023) and 7.8% larger than the $H_2$ GWP100 calculated here using present day atmospheric chemical composition. The atmospheric lifetime of hydrogen,

as well as the methane lifetime, methane feedback factor and $CH_4$ GWP100 depend on the chemical composition of the atmosphere. For hydrogen, however, the dominant factor for the total lifetime is the soil sink. A better understanding of the soil sink processes and how the soil sink may be affected by climate change is needed to further investigate how $H_2$ GWP100 may change in the future.

In conclusion, the $H_2$ GWP100 is independent of the size of the emission perturbation, depends to some degree on the

emission location (distance to soil sink active areas) and depends slightly on the chemical background atmosphere. Overall, these dependencies are small compared to the uncertainty in the $H_2$ GWP100 due to weaknesses in understanding of the processes controlling the hydrogen budget.

**Code availability**

The code to reproduce the figures in this manuscript can be found at https://zenodo.org/records/14823343 (Skeie et al.,

2025). The OsloCTM3 version used here are available at https://github.com/ciceroOslo/OsloCTM3/releases/tag/hydrogen-sensitivity-v1.

**Data availability**

The data needed to reproduce the figures in the manuscript can be found at https://zenodo.org/records/14823343 (Skeie et al., 2025). The full dataset for the model results is available in the NIRD Research Data Archive:

https://doi.org/10.11582/2025.00008 (Skeie, 2025).

**Author contribution**

RBS wrote the paper with contributions from all co-authors. Marit S designed the geographical experiments, RBS designed the SSP experiments. RBS performed the OsloCTM3 simulations. All co-authors discussed the design and results.

**Competing interests**

At least one of the (co-)authors is a member of the editorial board of Atmospheric Chemistry and Physics.

## Acknowledgements

The work has received funding from the Norwegian Research Council (80%) and and six industrial partners (20%): Shell, Equinor, Statkraft, Linde, Gassco and Norwegian Shipowners' Associations (HYDROGEN, grants no. 320240) and the European Union's Horizon Europe research and innovation programme under grant agreement no. 101137582 (HYway). The OsloCTM3 simulations were performed on resources provided by Sigma2 - the National Infrastructure for High-Performance Computing and Data Storage in Norway (project account NN9188K), and data uploaded and shared through their services (project NS11106K and NS9188K).

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
