# Peer review of "Sensitivity of climate effects of hydrogen to leakage size, location, and chemical background"

_EGUsphere, 2024_

## Referee Comment (RC1)

Comments for Paper entitled "Sensitivity of climate effects of hydrogen to leakage size, location, and chemical background"

This paper investigates the impacts on H2 and CH4 GWP100 due to changes in three different scenarios settings. These settings are : 1) changes in hydrogen emission perturbation, 2) pulse emissions of hydrogen at specific locations, and 3) three different SSP scenarios. They find that the H2 GWP100 does not depend on the magnitude of hydrogen emission perturbations. For specific locations of H2 emissions, the H2 GWP100 is different in locations far from soil uptake e.g. in the ocean and in Antarctica. For different SSP scenarios, the H2 GWP100 is dependent on both CH4 concentration and the NOx:CO ratio, both of which heavily influence OH and, by extension, H2 atmospheric lifetime and GWP100. Ultimately, however, the soil sink is the dominant driver factor for H2 soil rather than the H2 atmospheric lifetime. With the exception of the pulse experiments located in the ocean and Antarctica, all H2 GWP100 results are within one standard deviation of the GWP100 found in Sand et al. (2023).

This is a comprehensive study assessing the GWP100 of H2 under different situations. Hydrogen is an important topic in both in research and society and this is a valuable contribution to the hydrogen community in narrowing down the uncertainty of H2 and its impact on the climate.

**General comments**

 – Some of the results would benefit from further quantitative analysis to state whether these values are statistically significant (see comments below).

 – I think the authors need a further explanation as to why they've chosen their locations and to define what is meant by a "sink soil active area" as this is unclear to me.

– Given that the authors have described how interconnected the reactions are in the atmosphere, would be useful to have a forth experiment where both the H2 emissions and the CH4 concentration is enhanced. This would then clarify whether the enhancement due to H2 emissions and CH4 concentration is additive or if increasing both CH4 and H2 causes an further increase due to OH production/loss e.g. anthro1 for H2 and 10% increase in CH4 for present day. Author may have already taken this into consideration, in which case this should be clarified in the text

 – The conclusion of linearity between GWP100 and H2 emission perturbation is currently misleading and confusing, especially as authors later say these two variables are independent. From Fig 3 it is difficult to see how these are linear as well

**Specific comments**

Abstract
lines 9–10 : This sentence is vague – specify the reactions (OH induced) and what the effect on CH4, O3 and H2O are

Line 14 : See later comments about linearity of GWP100 + size of emission perturbation. Also, if it is linear, this is only true when you consider the magnitude of emission perturbation as the emission perturbations chosen are logarithmically increasing

Line 20 : This is a strong statement and as you point out in the conclusions it isn't taking into account soil sink. Can you add a clause that this is only considering OH sink of H2?

**Introduction**
line 25 : Specify how it will cause these greenhouse gases to change
line 44 : Missing refs. Either add more in, or give e.g.

**Methods**
line 75 : expand full model acronym at first instance

line 80 : Does the third experiment also have enhanced H2 emissions along with an enhanced CH4 concentration? Adding a sentence in to clarify would help this.

Line 86 : Merge into previous paragraph

Line 87 : Define what is meant by linearity here – if the emission perturbations are increasing by a magnitude of 10 in each experiment, "linearity" is misleading here unless you specifically refer to the logarithmic increase of perturbations

Line 87 : In line 79, the authors say they run 3 simulations to calculate GWP, and here they explain they have 3 sets of sensitivity tests (which include multiple simulations). I assume these are separate to the GWP runs described previously? Please could the authors add in a few sentences at the start of the methods section to summarise all the set of experiments they're doing more clearly. Authors might consider moving the "GWP Calculation" section to after the "Sensitivity experiments" (after Lines 103–109) have been explained to help with layout of explaining their experiment setup

Line 100 : "correspond to two different control simulations, as hydrogen is concentration driven in the methane perturbation simulation." Please expand on the differences between the control simulations.

Line 113 : "The OsloCTM3 model is used in a similar set up as in Sand et al. (2023)" Please give a brief description of what the set up is here.

Line  120 : Authors might consider using the corrected oceanic H2 emissions from Paulot et al. 2023 (Fig S3)

**Results**
Section 3.1: The title of this section is quite ambiguous at a first glance. Please could the authors rename it to be more informative

Line 163 : Similarly, in this sentence could the authors clarify "the results" (I assume they mean the H2 GWP100 values from the previous section?).

Line 165 : It's difficult to see how the GWP100 values have a linear relationship wrt magnitude of emission. The anthro100 GWP actually looks like it is lower than the anthro10. These values are very close together – can the authors authors say whether or not these are significantly different enough for it to be considered linear?

Line 178 : How do the authors define an soil sink active area? Is it based on the average soil uptake over an area?

Line 180 : Can the authors show these differences are statistically significant from one another? I think this would strengthen their argument

Line 185 : Referring to Fig. 1b at the end of this sentence is somewhat confusing as the numerical values (5.2 and 5.4) refer to Fig. 4a. The authors could move this Fig reference to a more suitable place in the sentence or leave it out entirely.

Line 195 : Could authors give an equation for the feedback factor than rather a word description and also define lifetime of perturbation for a complete explanation

Line 204 : Can the authors expand more on what is significant of the lowest feedback factor being 0.76 would mean in the larger context?

Line 201 : There doesn't look like there is much change in forcing if there is an increase in H2 burden from Fig 4. and all the GWP values are similar. Are these statistically significant values from each other to make this statement? Authors say that it is within the uncertainty range of values from Sand et al. 2023 so perhaps not?

Line 236 : Worth mentioning that the dominant chemical loss of H2 is via OH and/or refer to equation 1

Line 240–253 : This is nicely explained

Line 253: Can authors comment on the effectiveness of NOx: CO ratio and changes in CH4 of H2 lifetime?

Line 174 : Can authors suggest why the ozone contributes are greater in SSP119 than the others? Or is it that the ozone contribution is the same in all scenarios, but the contributions from strat. H2O and CH4 are lower in SSP126, resulting in a large proportion from ozone?

Line 297 : I might be misunderstanding this, but earlier in lines 163–6 authors say results are linear (which is unconvincing from the graph), but also say they are independent with respect to the magnitude of emission perturbation. These two conclusions seem mutually exclusive. Can authors describe what they mean by "independent" of hydrogen emission perturbation, but have a "linear response"?

Line 306 : Can authors clarify how the longitudinal dependence links to soil sink active areas (e.g. inland vs by the coast?)

Line 315 : Given that the authors are looking at particular locations, it's not surprising that GWP values are outside the standard deviation values from Sand et al., especially as these are at the extremes which won't be captured in a standard deviation range. Consider rephrasing the sentence 314–316 so it is less defensive of this result!

Line 335 : Rephrase to "lifetime decreased by a range of 0.19–1.1 years" or equivalent

Line 368–9 : Can authors rephrase this sentence to include quantitative values to support their argument?

Line 370–1 : "due to process understanding of the hydrogen budget." I assume the authors specifically mean the soil sink budget?

Table 2 : State what the starting concentration is for CH4 concentration in present–day without 10% increase

Fig 1b : Could the authors also provide a figure of soil deposition in units commonly used in other papers e.g. cm s–1 so it can be compared to other soil deposition models (e.g. Paulot et al. 2021, Bertagni et al. 2023).

Fig 2a: I assume the authors mean the surface methane concentration – can you add this into the caption for clarity

Fig 3: Are these values averaged over one year or multiple years?

**Technical comments**

line 56 : NOx vs $NO_x$

line 68 : "as well *as*"

line 86 : Combine sentence with previous paragraph

line 87 : First sentence is a repeat from paragraph before – remove

Fig 1b: Move the caption or change the colour in the high northern latitudes as it is difficult to read

Table 1: missing "s" in "magnitude of the emission perturbation"

line 276 : Missing % after 3.0

line 296 : No need to reference Sand et al. At the end given the start of the sentence

line 360 : $NO_x$ vs NOx

line 370 : process → processes

---

## Author Response (AR1)

We would like to thank the two reviewers for their useful comments to this manuscript. Below follows our responses to the comments by the reviewers and a description of how the manuscript has been modified. The original reviewer's comments are in green and our response in black.

**Referee #1**

**Comments for Paper entitled "Sensitivity of climate effects of hydrogen to leakage size, location, and chemical background"**

This paper investigates the impacts on H2 and CH4 GWP100 due to changes in three different scenarios settings. These settings are : 1) changes in hydrogen emission perturbation, 2) pulse emissions of hydrogen at specific locations, and 3) three different SSP scenarios. They find that the H2 GWP100 does not depend on the magnitude of hydrogen emission perturbations. For specific locations of H2 emissions, the H2 GWP100 is different in locations far from soil uptake e.g. in the ocean and in Antarctica. For different SSP scenarios, the H2 GWP100 is dependent on both CH4 concentration and the NOx:CO ratio, both of which heavily influence OH and, by extension, H2 atmospheric lifetime and GWP100. Ultimately, however, the soil sink is the dominant driver factor for H2 soil rather than the H2 atmospheric lifetime. With the exception of the pulse experiments located in the ocean and Antarctica, all H2 GWP100 results are within one standard deviation of the GWP100 found in Sand et al. (2023).

This is a comprehensive study assessing the GWP100 of H2 under different situations. Hydrogen is an important topic in both in research and society and this is a valuable contribution to the hydrogen community in narrowing down the uncertainty of H2 and its impact on the climate.

**General comments**

- Some of the results would benefit from further quantitative analysis to state whether these values are statistically significant (see comments below).

See our response to this comment below.

- I think the authors need a further explanation as to why they've chosen their locations and to define what is meant by a "sink soil active area" as this is unclear to me.

We have modified the text to clarify why we have chosen the different locations. "Soil sink active area" as defined here was areas with the largest amount of H2 deposited over a year. Using this definition, it is also dependent on where the H2 is emitted, and in retrospect we should have used the dry deposition velocities instead. We have decided to replace Fig. 1b with a figure showing the dry deposition velocities. The chosen locations fit well. There is an area in the US where the deposition velocities are large. For

the maxdep site, the chosen grid cell is not the grid cell with the largest deposition velocity. The gridcell with the largest deposition velocity is in the north eastern corner of the Caspian sea (an area with very little emissions and hence low total deposition). This point is at the same latitude as epia, and may therefore not be so interesting. It is more interesting to look at a site at lower latitudes with large soil sink. We rename the "maxdep" site to "lowlatdep" as it is located at low latitudes in a soil sink active area.

In the method section we have now written:

"*a point in the US in an area with large soil sink velocities in the model (usdrydep); a point in East Africa close to the Equator where the soil sink velocity is large (lowlatdep);*"

[Figure]

**Figure 1: In a) the total annual anthropogenic emissions of hydrogen from Paulot et al. (2021) for 2010 is shown and in b) the annual mean deposition velocity of hydrogen from OsloCTM3 present-day control simulation is shown. In both**

**figures, the sites where 1 Tg yr$^{-1}$ is added in the sensitivity tests for perturbation locations are indicated by a blue star and in b) the names of the respective sensitivity tests are also indicated.**

- Given that the authors have described how interconnected the reactions are in the atmosphere, would be useful to have a forth experiment where both the H2 emissions and the CH4 concentration is enhanced. This would then clarify whether the enhancement due to H2 emissions and CH4 concentration is additive or if increasing both CH4 and H2 causes an further increase due to OH production/loss e.g. anthro1 for H2 and 10% increase in CH4 for present day. Author may have already taken this into consideration, in which case this should be clarified in the text

A next step in assessing the atmospheric impact of H$_2$ emissions, is to do the hydrogen perturbation in a simulation with free running methane. When methane also is emission driven, only one perturbation simulation is needed. Then we can compare the results from that simulation with the method performed here, combining the two simulations (hydrogen perturbation and methane perturbation). That will be the approach to see if the enhancement of H2 and CH4 is additive. This is beyond the scope of this sensitivity study but simulations of this are planned in an ongoing EU project (HYway).

- The conclusion of linearity between GWP100 and H2 emission perturbation is currently misleading and confusing, especially as authors later say these two variables are independent. From Fig 3 it is difficult to see how these are linear as well

There are some confusions related to the linearity in the H$_2$ emissions and GWP. As suggested by the other reviewers, we have tried to clarify and e.g. renamed the section 3.1 to "Is the response to emissions size linear?"

And in the abstract we have written:

"*We show that the hydrogen GWP100 is independent of the size of the emission perturbation,*"

**Specific comments**

**Abstract**

lines 9-10 : This sentence is vague – specify the reactions (OH induced) and what the effect on CH4, O3 and H2O are

Due to word limitations of the abstract we have only slightly modified this sentence to make it more specific:

*"Via OH-induced reactions in the atmosphere, the hydrogen will enhance methane, ozone, and stratospheric water vapor in the atmosphere and hence increase the radiation imbalance."*

Line 14 : See later comments about linearity of GWP100 + size of emission perturbation. Also, if it is linear, this is only true when you consider the magnitude of emission perturbation as the emission perturbations chosen are logarithmically increasing

See response below. And indeed the linearity is tested only for the range of 0.1 to 100 Tg yr$^{-1}$.

Line 20 : This is a strong statement and as you point out in the conclusions it isn't taking into account soil sink. Can you add a clause that this is only considering OH sink of H2?

This statement is rewritten as:

*"Therefore, when assessing emissions at different locations or for a future with different atmospheric composition than present-day, it is not necessary to adjust the multi-model GWP values."*

**Introduction**

line 25 : Specify how it will cause these greenhouse gases to change

We have specified that it reacts with OH. "*The leaked hydrogen will react in the atmosphere with the hydroxyl radical (OH) and alter the abundance of other greenhouse gases:*" The reaction is also presented later in the introduction.

line 44 : Missing refs. Either add more in, or give e.g.

We have added an e.g. here.

**Methods**

line 75 : expand full model acronym at first instance

We have added that it is a global chemical transport model: "We use the global chemical transport model OsloCTM3"

line 80 : Does the third experiment also have enhanced H2 emissions along with an enhanced CH4 concentration? Adding a sentence in to clarify would help this.

No, this is a pure methene perturbation experiment.

This is clarified: "*From the methane perturbation experiment, where hydrogen is the same as in the control simulation, ...*"

Line 86 : Merge into previous paragraph

We have merged this sentence into the previous paragraph.

Line 87 : Define what is meant by linearity here – if the emission perturbations are increasing by a magnitude of 10 in each experiment, "linearity" is misleading here unless you specifically refer to the logarithmic increase of perturbations

We have rewritten this part as: "*In the first set of sensitivity tests, we test if the calculated GWP100 is dependent on the size of the emission perturbation.*" As there seems to be confusion about what linearity means. The test is to see if the response of the hydrogen perturbation is linear with respect to the size of the perturbation. See also response to other comments.

Line 87 : In line 79, the authors say they run 3 simulations to calculate GWP, and here they explain they have 3 sets of sensitivity tests (which include multiple simulations). I assume these are separate to the GWP runs described previously? Please could the authors add in a few sentences at the start of the methods section to summarise all the set of experiments they're doing more clearly. Authors might consider moving the "GWP Calculation" section to after the "Sensitivity experiments" (after Lines 103-109) have been explained to help with layout of explaining their experiment setup

The section 2.1 GWP calculation is rewritten and the list of 3 simulations is now removed so that it will not be confused with the three set of sensitivity tests. At the end of the GWP section we have included the following:

"We follow the same approach as in the multi-model study by Sand et al. (2023). While the hydrogen concentration was perturbed in the main simulations in that study, we use hydrogen emission perturbations and investigate the sensitivity of the GWP100 to how the perturbations are implemented in the model."

Before describing the different sensitivity experiments in section 2.2, we think it is useful to first describe how GWP is calculated.

Line 100 : "correspond to two different control simulations, as hydrogen is concentration driven in the methane perturbation simulation." Please expand on the differences between the control simulations.

We have added that hydrogen is emission driven in the hydrogen perturbation simulations:

"*Note that the methane perturbation and the sensitivity tests for hydrogen perturbations correspond to two different control simulations, as hydrogen is concentration driven in*

*the methane perturbation simulation and emission driven in the hydrogen perturbation simulations.*"

Line 113 : "The OsloCTM3 model is used in a similar set up as in Sand et al. (2023)" Please give a brief description of what the set up is here.

We have added a brief description of the exceptions to the setup as used in Sand et al (see response to the other reviewer). The setup including meteorological data and years, simulation length, resolution, emissions are all described in this paragraph.

Line 120 : Authors might consider using the corrected oceanic H2 emissions from Paulot et al. 2023 (Fig S3)

Thank you for pointing out these revised emissions in Paulot et al 2024 (I think you mean). It is not feasible to redo all the simulations in this study at this stage, and we do not think slightly altering the background hydrogen emissions and shifting the geographical distributions of the oceanic H2 emissions are important for the sensitivity results. There are large uncertainties in all the H2 budget terms, and that is highlighted at the end of the manuscript:

"*Overall, these dependencies are small compared to the uncertainty in the $H_2$ GWP100 due to weaknesses in understanding of the processes controlling the hydrogen budget.*"

**Results**

Section 3.1: The title of this section is quite ambiguous at a first glance. Please could the authors rename it to be more informative

As suggested by the other reviewer, the section is renamed:

"*Is the response to emissions size linear?*"

Line 163 : Similarly, in this sentence could the authors clarify "the results" (I assume they mean the H2 GWP100 values from the previous section?).

We have rewritten this section, hopefully the linearity confusion is solved.

"*The first set of sensitivity tests investigate the linearity in the chemical response with respect to the magnitude of the hydrogen emission perturbation. Adding a hydrogen perturbation of 0.1, 1, 10 or 100 Tg yr$^{-1}$ resulted in very similar GWP100 values ranging from 12.2 to 12.6 as seen in the first 4 bars in Fig. 3. To conclude, the GWP100 values are independent of the magnitude of the emission perturbation within the range 0.1 to 100 Tg yr$^{-1}$.*"

Line 165 : It's difficult to see how the GWP100 values have a linear relationship wrt magnitude of emission. The anthro100 GWP actually looks like it is lower than the

anthro10. These values are very close together – can the authors authors say whether or not these are significantly different enough for it to be considered linear?

There is some misunderstanding here. It is the response to the emission perturbation that is linear. As GWP is normalized by emission, a linear response means the GWP is similar regardless of emission size. And these values are very similar indeed. This has been clarified in the manuscript (see also other review response comments).

Line 178 : How do the authors define an soil sink active area? Is it based on the average soil uptake over an area?

Soil sink active area is where $H_2$ is deposited in the model. We hope, by replacing Fig. 1b with the deposition velocities from the model this is clearer from the Method section. We have also included a reference to Fig. 1b here.

Line 180 : Can the authors show these differences are statistically significant from one another? I think this would strengthen their argument

We are not able to calculate the statistically significance of these results, as we have used a single chemical transport model, and there is no variability in the results.

However, taking the uncertainties in the GWP calculations into account, as was done in Sand et al. 2023, the variability in the results here is mostly within the 1 std. deviation. The reason for including the Sand et al bar in the figure is indeed to show that the results from the sensitivity tests performed here have small variability compared to the full uncertainty. We end this section by comparing the results to the uncertainty estimate in Sand et al.

Line 185 : Referring to Fig. 1b at the end of this sentence is somewhat confusing as the numerical values (5.2 and 5.4) refer to Fig. 4a. The authors could move this Fig reference to a more suitable place in the sentence or leave it out entirely.

Yes, the reference to Fig. 1b is not suited here and is therefore deleted.

Line 195 : Could authors give an equation for the feedback factor than rather a word description and also define lifetime of perturbation for a complete explanation

This section is rewritten based on the other reviewer's comment, and the focus is on the perturbation lifetime now. For methane we still use the chemical feedback term, and have written the following: "*For methane the chemical feedback factor (the lifetime of the perturbation divided by the background lifetime of the atmospheric component) is larger than 1…*".

Line 204 : Can the authors expand more on what is significant of the lowest feedback factor being 0.76 would mean in the larger context?

As we have shifted the focus from feedback factor to perturbation lifetime the following part should put the results in the larger context "*The longer the lifetime of the hydrogen perturbation, the larger the change in hydrogen burden in the atmosphere. A larger change in hydrogen burden leads to a larger forcing and forcing per flux of hydrogen (as shown in Fig. 4b for CH$_4$, Fig. 4c for O$_3$, and Fig. 4d for strat. H$_2$O) and larger GWP100. Similarly, a smaller change in atmospheric hydrogen per hydrogen flux leads to smaller forcing and smaller GWP100 values.*"

Line 201 : There doesn't look like there is much change in forcing if there is an increase in H2 burden from Fig 4. and all the GWP values are similar. Are these statistically significant values from each other to make this statement? Authors say that it is within the uncertainty range of values from Sand et al. 2023 so perhaps not?

Yes, as the results are within the one standard deviation of Sand et al., they are not expected to be statistically significant. As explained above, the results here are not suitable for significance testing.

Line 236 : Worth mentioning that the dominant chemical loss of H2 is via OH and/or refer to equation 1

We have added: "the chemical loss through OH (Eq. 1),"

Line 240-253 : This is nicely explained

Thank you!

Line 253: Can authors comment on the effectiveness of NOx: CO ratio and changes in CH4 of H2 lifetime?

As the SSP simulations change both NOx:CO ratio and CH4, it is difficult to separate the effectiveness of the two on the H2 lifetime. We see that in SSP119 and SSP585 the change in NOx:CO and CH4 push in different direction with respect to OH, where the NOx:CO ratio dominates in SSP119 and CH4 dominate in SSP585. Separate idealized simulations would be needed to estimate the effectiveness of NOx:CO ratio and changes in CH4 in the atmospheric lifetime of H2.

Line 174 : Can authors suggest why the ozone contributes are greater in SSP119 than the others? Or is it that the ozone contribution is the same in all scenarios, but the contributions from strat. H2O and CH4 are lower in SSP126, resulting in a large proportion from ozone?

As shown in Fig. 4c, and described in the text, the direct contribution from ozone is larger in SSP119 compared to the two other scenarios.

Why the ozone contribution is greater in SSP119 than in the other is not easily explained. Ozone precursor emissions are different in the scenarios and compared to the present day (Fig. S1), and the non-linearity in the ozone chemistry plays a part.

Line 297 : I might be misunderstanding this, but earlier in lines 163-6 authors say results are linear (which is unconvincing from the graph), but also say they are independent with respect to the magnitude of emission perturbation. These two conclusions seem mutually exclusive. Can authors describe what they mean by "independent" of hydrogen emission perturbation, but have a "linear response"?

See other replies to related comments.

Line 306 : Can authors clarify how the longitudinal dependence links to soil sink active areas (e.g. inland vs by the coast?)

Fig. 1b, that now shows the dry deposition velocities, shows where the soil sink is most active. Although we have a limited number of tests, we see that usdrydep, epia and munich (in approximately the same latitude band) show some differences in the GWP values. We hope this is clearer now with the rewritten method section.

Line 315 : Given that the authors are looking at particular locations, it's not surprising that GWP values are outside the standard deviation values from Sand et al., especially as these are at the extremes which won't be captured in a standard deviation range. Consider rephrasing the sentence 314-316 so it is less defensive of this result!

314-316: *"The GWP100 values span the one standard deviation range from Sand et al. (2023), but note that some of the sites chosen are remote locations (in the Arctic, Antarctic, Southern Ocean) that are not relevant locations for future hydrogen economy."*

We have added the following: *"For continental sites more relevant for a future hydrogen economy, the GWP100 values are well within the one standard deviation range assessed in Sand et al. (2023)."*

Line 335 : Rephrase to "lifetime decreased by a range of 0.19-1.1 years" or equivalent

As suggested, we have rephrased this.

Line 368-9 : Can authors rephrase this sentence to include quantitative values to support their argument?

This comment refers to the second to last sentence in the manuscript "In conclusion, the H2 GWP100 is independent of the size of the emission perturbation, depends to some degree on the emission location (distance to soil sink active areas) and depends slightly on the chemical background atmosphere." Previous in the conclusion section we have quantified these numbers; 10.2 to 14.2 for the perturbation location experiments, now added *"where the values range from 12.2 to 12.6"* for the test on the emission perturbation size. We have also added that the GWP100 is 13.5 in SSP585. The quantitative values to this argument are hence given in the previous paragraphs of this conclusion section.

Line 370-1 : "due to process understanding of the hydrogen budget." I assume the authors specifically mean the soil sink budget?

Yes, especially the soil sink part of the budget. But also the other parts of the budget are uncertain. If the rest of the budget terms would have been better represented, the soil sink term would have been known as well.

Table 2 : State what the starting concentration is for CH4 concentration in present-day without 10% increase

The number in the parentheses is the starting concentration without the 10% increase.

We have removed the parentheses and instead clearly written: "*from XXXX ppbv in the control*"

Fig 1b : Could the authors also provide a figure of soil deposition in units commonly used in other papers e.g. cm s-1 so it can be compared to other soil deposition models (e.g. Paulot et al. 2021, Bertagni et al. 2023).

We have replaced the deposition figure with deposition velocity instead (see also response to other review comments). Hopefully this will clarify what soil sink active regions are.

Fig 2a: I assume the authors mean the surface methane concentration – can you add this into the caption for clarity

We have added "surface" to the caption.

Fig 3: Are these values averaged over one year or multiple years?

These values are for the last year of the simulations. As we have used a CTM, driven by the same meteorological fields, there is no interannual variability confounding the results. We have specified this in the results section (section 2.4): "*The ERFs for these components are calculated from the last year of the OsloCTM3 steady-state simulations.*"

**Technical comments**

line 56 : NOx vs $NO_x$

Done.

line 68 : "as well *as*"

Done.

line 86 : Combine sentence with previous paragraph

Done.

line 87 : First sentence is a repeat from paragraph before – remove

I think the three simulations needed to calculate the GWP and the three sets of sensitivity experiments are mixed here. We have therefore removed the numbering from the paragraph before.

Fig 1b: Move the caption or change the colour in the high northern latitudes as it is difficult to read

We have moved the text on this figure to improve readability.

Table 1: missing "s" in "magnitude of the emission perturbation"

Corrected.

line 276 : Missing % after 3.0

Added.

line 296 : No need to reference Sand et al. At the end given the start of the sentence

We have deleted the references to Sand et al here to make the manuscript more independent of that study.

line 360 : NO$_x$ vs NOx

We have replaced all NOx with NO$_x$ throughout the manuscript.

line 370 : process → processes

Replaced.

**Referee #2**

This manuscript describes an assessment of the climate impacts of hydrogen leakages and the sensitivity of these to emissions magnitude and location under different future scenarios for atmospheric composition. The study concludes that the climate impacts are largely independent of the magnitude of emission changes, that there is some dependence on location (although this is small for expected emission locations), and that the impacts will change under different future atmospheric composition, but by only a small margin. This is the first time that these effects have been explored in a consistent way in a global chemistry transport model, and the results are valuable, giving confidence to previous assessments of global warming potential. The methodology used is sound, although there are a number of places where it could be described more clearly. I have a number of minor concerns (detailed below), but in other respects this is an interesting and valuable paper, and merits publication in ACP once the following issues have been addressed.

**General Comments**

The paper refers heavily to the study of Sand et al 2023 and is largely presented as an extension of this previous work. While reference to this useful study is certainly merited, the paper would be stronger if presented in a more independent manner so that the reader does not feel that they must read the earlier study first. Frequent reference to the Sand et al. paper detracts from the novelty of this work.

Thank you for pointing this out, we agree with your assessment and have tried to adjust accordingly.

At the end of the introduction section, we refer to more studies than Sand et al:

*"In this study we test the robustness of the GWP100 results presented in previous studies where hydrogen concentrations or hydrogen emissions are enhanced globally (Warwick et al., 2023;Hauglustaine et al., 2022;Sand et al., 2023)."*

Especially in the Method section, the text is slightly rewritten and references to Sand et al. removed. In the Method section we refer to Sand et al first at the end of section 2.1:

*"We follow the same approach as in the multi-model study by Sand et al. (2023). While the hydrogen concentration was perturbed in the main simulations in that study, we use hydrogen emission perturbations and investigate the sensitivity of the GWP100 to how the perturbations are implemented in the model."*

The differences in GWP due to location are attributed to the H2 soil sink. While this effect is likely to be dominant, the abundance of OH in these locations also differs substantially. What are the relative contributions of these two sinks for the applied perturbations? No evidence is presented that the soil sink is the sole cause of the differences.

We have added soil sink lifetime and perturbation lifetime as well as atmospheric lifetime and perturbation lifetime to Table 3 to highlight this point. We have also rewritten the text as a response to your next comment (see below).

Table 3: Total lifetime of hydrogen (total burden divided by the total loss in the control simulation), atmospheric lifetime (total burden divided by the atmospheric loss in the control simulation) and soil sink lifetime (total burden divided by the soil sink loss) and the respective perturbation lifetimes (change in burden divided by change in loss in the perturbation simulation relative to the control simulation) for the different sensitivity tests. The change in perturbation lifetime relative to the lifetime in the control simulation (in %) are given in the parentheses.

| | H$_2$ total lifetime [yrs] | H$_2$ perturbation lifetime [yrs] | H$_2$ atmospheric lifetime [yrs] | H$_2$ atmospheric perturbation lifetime [yrs] | H$_2$ soil sink lifetime [yrs] | H$_2$ soil sink perturbation lifetime [yrs] |
|---|---|---|---|---|---|---|
| anthro01 | 2.35 | 2.17 (-7.5) | 7.02 | 7.23 (3.1) | 3.53 | 3.10 (-12) |
| anthro1 | 2.35 | 2.17 (-7.5) | 7.02 | 7.24 (3.2) | 3.53 | 3.10 (-12) |
| anthro10 | 2.35 | 2.17 (-7.4) | 7.02 | 7.27 (3.6) | 3.53 | 3.10 (-12) |
| anthro100 | 2.35 | 2.20 (-6.3) | 7.02 | 7.55 (7.6) | 3.53 | 3.10 (-12) |
| nemo | 2.35 | 2.56 (9.0) | 7.02 | 7.64 (8.9) | 3.53 | 3.85 (9) |
| epia | 2.35 | 1.94 (-17.2) | 7.02 | 7.25 (3.3) | 3.53 | 2.66 (-25) |
| munich | 2.35 | 2.08 (-11.3) | 7.02 | 7.24 (3.2) | 3.53 | 2.92 (-17) |
| usdrydep | 2.35 | 1.77 (-24.5) | 7.02 | 7.18 (2.3) | 3.53 | 2.35 (-33) |
| maud | 2.35 | 2.59 (10.2) | 7.02 | 7.73 (10.2) | 3.53 | 3.89 (10) |
| zep | 2.35 | 2.09 (-11.0) | 7.02 | 7.40 (5.5) | 3.53 | 2.91 (-17) |
| lowlatdep | 2.35 | 1.99 (-15.0) | 7.02 | 7.22 (2.9) | 3.53 | 2.75 (-22) |
| SSP119 | 2.37 | 2.20 (-6.9) | 7.26 | 7.59 (4.6) | 3.51 | 3.11 (-12) |
| SSP434 | 2.45 | 2.26 (-7.8) | 7.95 | 8.26 (3.9) | 3.54 | 3.11 (-12) |
| SSP585 | 2.41 | 2.22 (-8.0) | 7.55 | 7.80 (3.3) | 3.55 | 3.11 (-13) |

The term "feedback factor" is not used correctly here. Methane is sufficiently long-lived to be well-mixed in the troposphere, and is a substantial sink of OH, and therefore has a chemical feedback on its own lifetime. A methane perturbation thus decays more slowly than expected as global OH is perturbed. Hydrogen has a similar, but smaller effect on OH, and therefore has a feedback factor slightly greater than one for a well-mixed global perturbation. The different perturbation lifetimes for hydrogen identified here are due to differences in distribution (and thus to the relative balance of sinks) and not to a specific feedback process. The soil sink, in particular, is first-order, so no feedback is possible. While the tropospheric mean lifetime will increase or decrease under different perturbations, depending on location, it is misleading to describe this as a feedback factor. The perturbation lifetime is simply different from the global lifetime, as is true for most reactive gases. Different terminology is needed here.

We have rewritten the section and removed the term "feedback factor" from the text and Table 3, and instead focus on the perturbation lifetime.

From line 196, we have rewritten the section:

"*The difference in the increase of atmospheric hydrogen concentration per flux of hydrogen can be explained by different perturbation lifetimes. For chemically reactive*

*species, such as methane, the chemical loss in the atmosphere via OH will be less efficient when more methane is added to the atmosphere, and the lifetime of the methane perturbation is enhanced. For methane the chemical feedback factor (the lifetime of the perturbation divided by the background lifetime of the atmospheric component) is larger than 1 (Holmes et al., 2013;Sand et al., 2023;Thornhill et al., 2020). For a hydrogen perturbation, the geographical distribution of the emission perturbation influences the perturbation lifetime (Table 3). For the atmospheric lifetime, the perturbation lifetime is larger than the lifetime, with the largest increase of 10% at the Antarctic site (maud). For the soil sink lifetime, the perturbation lifetime is shorter than the hydrogen background lifetime except for the ocean site (nemo) and Antarctic site (maud) (Table 3). The soil sink is enhanced when emissions are close to the soil sink active areas relative to the total soil sink and hence the perturbation lifetime is reduced. As the soil sink lifetime (3.5 years) is lower than the atmospheric lifetime (7.0 years) a similar change is seen for the total perturbation lifetime relative to the hydrogen lifetime as for the soil sink lifetime (Table 3). The largest difference between the perturbation lifetime and the hydrogen lifetime is for usdrydep with a difference of 0.57 years.*"

In the discussion the relevant text is rewritten as follows:

"*The reason for the different GWP values in the geographical sensitivity tests can be explained by differences in the perturbation lifetime. For the sites nemo and maud, the perturbation lifetime is enhanced relative to the lifetime of hydrogen in the control simulation. For sites close to areas where hydrogen can be taken up by the soil, the perturbation lifetime is shorter than the hydrogen lifetime. This is because the soil sink is enhanced when emissions are close to the soil sink active areas relative to the total soil sink. In Sand et al. (2023) the perturbation lifetime relative to the background lifetime is slightly less than one, and ranges from 0.95 to 1.0. These simulations were mostly concentration driven and hydrogen concentration was enhanced by 10% globally. For the geographical sensitivity tests, the range in perturbation lifetime was from 1.8 to 2.6 years, 25% shorter to 10% longer than the background lifetime. The GWP100 (10.2 to 14.2) values span the one standard deviation range from Sand et al. (2023)...*"

The paper notes that the locations with the largest GWP100 values are not very relevant for the future hydrogen economy. What conclusions can you draw from the sites that are most relevant for the hydrogen economy? If future emissions increase in populated continental regions, will GWP100 be lower than estimates for current conditions?

We have added the following in the discussion section:

"*For continental sites more relevant for a future hydrogen economy, the GWP100 values are well within the one standard deviation range assessed in Sand et al. (2023).*"

And similarly at the end of section 3.2 to highlight the results from the more relevant continental sites:

"*For the continental sites more relevant for a future hydrogen economy (usdrydep, munich, lowlatdep, epia) the GWP100 range from 10.2 to 11.9, slightly less than perturbing the total anthropogenic emissions (anthro1) of 12.5.*

**Specific Comments**

Line 23: "only water vapour is emitted": this is not true if hydrogen is burned, so please rephrase.

Yes, when hydrogen is directly burned NOx emissions will also occur, but used in a fuel cell there will be no NOx emissions. We have rewritten the sentence as the following:

"*Hydrogen is not a greenhouse gas and when it is used in a fuel cell to generate energy, only water vapor is emitted.*"

Line 74: "steady-state perturbation approach" is not immediately clear to readers who haven't read the Sands et al. paper. It would be clearer to use "emissions perturbation approach" here and then explain the steady state aspect in the next paragraph.

We have rewritten the introduction to the method section as well as the beginning of section 2.1, also to make the manuscript less dependent on the Sand et al study.

"*In this study we use an emission perturbation approach to calculate the GWP100 of hydrogen. We use the global chemical transport model OsloCTM3 to investigate the sensitivity of the calculated GWP100 due to the size and location of the hydrogen perturbation, as well as future atmospheric chemical composition.*"

In section 2.1, we take one step back and introduce the steady-state perturbation approach:

"*The $H_2$ GWP is the ratio of the absolute global warming potential (AGWP) for hydrogen relative to that for $CO_2$. The AGWP is defined as the time integrated effective radiative forcing of a 1 kg pulse emission over a given time horizon (Myhre et al., 2013). For a 100 year time horizon all the perturbations from an initial hydrogen pulse have decayed, and it is shown that a steady state perturbation matches the integrated response of a pulse emission (Prather, 2002;Prather, 2007).*"

Line 82: please explain why the methane concentration is fixed at the surface, and note that this prevents it responding to the changes in OH due to hydrogen. The reader needs to appreciate this to understand why the third simulation is needed. Is the enhanced methane concentration in addition to the enhanced hydrogen emissions, or instead of them? Explain how the methane adjustment is made (or are these all +10%?)

We have tried to clarify this in the text now:

"*To calculate the H₂ GWP100, a control simulation and a simulation with enhanced hydrogen emissions are run to steady state. From the perturbed hydrogen simulation, the change in atmospheric composition of $O_3$, $CH_4$, and strat. $H_2O$ and the resulting ERFs due to these changes are calculated. As emission driven simulations of methane are challenging due to large uncertainties in the methane sources and sinks* sinks *(Saunois et al., 2020), global chemical models fix the surface concentration of methane. The change in the atmospheric methane is calculated from the modeled change in methane lifetime. As changes in methane also change the composition of $O_3$ and strat. $H_2O$ a methane perturbation experiment is also needed. From the methane perturbation experiment, where hydrogen is the same as in the control simulation, the atmospheric composition changes of $O_3$ and strat. $H_2O$, and hence the ERF, due to the changes in the methane lifetime in the hydrogen perturbation experiment can be extracted. The contributions to the H₂ GWP100 from the changes in the methane lifetime are referred to as "methane induced". From the methane perturbation experiments performed, we also calculate the $CH_4$ GWP100.*"

Line 93: please introduce reader to the seven locations before referring to them (perhaps just reference Fig 1 at end of the previous sentence).

We have added a reference to Fig. 1 in the previous sentence.

Lines 114-115: "has been updated", "has been corrected": please explain how (briefly) so that the reader can understand the changes.

We have explained this briefly:

"*... stratospheric water vapor has been updated to ensure a balanced hydrogen budget in the stratosphere and the geographical distribution of the anthropogenic emissions of hydrogen that was shifted 180 degrees has been corrected.*"

Line 119: Which version of GFED? version 4?

Yes this is version 4. Added GFED4s before the reference to the paper describing these emissions.

Line 129: please explain (briefly) what the adjustment term is for.

We have added:

"*, where the adjustment term converts the forcing from stratospheric temperature adjusted radiative forcing to effective radiative forcing.*"

Fig 1: Please adjust the position of the site labels so that they are legible. Given that the units are the same, why are the emissions presented on a log scale but the deposition on a linear scale? This is misleading and makes the panels difficult to compare.

We have adjusted the site labels so they are easier to read. The point of this figure was not to compare the a) and b), but in a) to show the field of emissions that was enhanced in the different anthro tests and in b) the different locations relative to where the model deposit $H_2$.

We will replace Fig. 1b with deposition velocity instead (see comment to reviewer 1) and the units are no longer the same.

Line 162: "Are the results linear" is unclear as a subtitle given the different sets of simulations described. Please use "Is the response to emissions size linear?" (or something similar).  The conclusion of this paragraph is that the response is linear, but yet the results differ slightly; what is the uncertainty in the GWP calculation?  The CH4 response starts to introduce nonlinearity at higher H2 (evident in Fig 4b), so large H2 emissions (1000 Tg/yr) are expected to cause the GWP to fall. The response is therefore only linear up to a certain emission size. Please make this clear on line 166.

We have renamed the subsection title as you suggested. In the concluding sentence, we have added that this result is valid for emission perturbations within the range of 0.1 to 100 Tg yr$^{-1}$.

"*To conclude, the GWP100 values are independent of the magnitude of the emission perturbation within the range 0.1 to 100 Tg yr$^{-1}$*"

Line 181: global mean surface hydrogen concentration?

Yes, added "global mean".

Line 195: "lifetime of the atmospheric component" isn't clear here; should this be the steady-state lifetime?

This part of the manuscript is rewritten (see response above). This lifetime is the lifetime in the control simulation.

Line 227: Note here that the chemical loss of hydrogen is also reduced (as described in the next paragraph), but this effect is smaller, so the changes in chemical production dominate the overall change.

We have added the word "mainly" as also the lifetime increases (and the chemical loss is reduced).

Table 5: The methane lifetime here is very short; the observation-based estimate is about 9.1 years, based on a chemical lifetime to OH of 11.2 years (Prather et al). This

indicates that OH concentrations are too high. What are the likely impacts of this on the GWP estimates presented for hydrogen?

The methane lifetime is short compared to observation-based estimates and hence OH is too high. It is hard to predict how a reduction in OH will impact the GWP100 of hydrogen. If OH is lower, the atmospheric lifetime of hydrogen will increase (GWP100 will increase, but the lifetime is dominated by soil sink, so probably small change), the ozone chemistry will be affected (due to nonlinear chemistry hard to predict the response), and the relative change in methane lifetime will be larger. The simulations here all had an increase in OH, but small change in GWP100, and in SSP119 and SSP434 the ozone contribution and methane contribution to the GWP100 were compensating for each other.

For the H2 GWP, the uncertainty in the soil sink is the dominant control on the lifetime and the main contributor to the uncertainty in the H2 GWP100. We will note in the second to last paragraph in the Discussion section the uncertainty due to the large OH levels in this study.

"*As noted above, the modelled methane lifetime is short and hence the OH level is high. Lower OH levels will enhance the lifetime of methane and the CH$_4$ GWP values but also impact the atmospheric chemistry involving ozone production. Further studies should investigate the role of OH levels on both H$_2$ and CH$_4$ GWP100.*"

In the start of the next paragraph, we highlighted that the soil sink is the dominant control of the total lifetime of hydrogen.

Line 279: These changes in methane feedback factor look very large. Please place these in the context of results from other studies in the literature (this is done later on lines 323-327, but the implications for the results presented here remain unclear).

In the discussion section, where we list results from other studies, we have tried to put our results in context and we have added: "*As methane lifetime is low in this model (Table 5) compared to assessed estimates of 9.1 ± 0.9 years (Szopa et al., 2021), the resulting methane feedback factor and chemical response following a perturbation, may be different in a model with lower OH levels and longer methane lifetime.*"

Line 296: "within one model": the model used in this study, or a different one?

Based on the comment to make the manuscript less dependent on the Sand et al study, we have deleted the references to Sand et al paper here.

The introduction to the discussion section is now as follows:

*"In this study, we have used a chemical transport model to investigate the sensitivity of the $H_2$ GWP100 on how the hydrogen emission perturbation is added in the model simulations."*

Line 297: As noted above, a caveat is needed on the conclusion that GWP100 is independent of emission perturbation size, as only small (realistic?) changes have been explored in this study.

We have added *"in the range of 0.1 to 100 Tg yr$^{-1}$ explored in this sensitivity study."*

All the location differences are attributed to the soil sink, but what about proximity to high OH regions? This is not mentioned in the manuscript.

In Table 3 we have now added atmospheric lifetimes and soil sink lifetimes. As soil sink is the dominant sink, this is the main reason for the differences in the GWP values. See response to comment above, where Table 3 results are discussed.

**Technical Corrections**

Line 13: "perturbation" not needed

Deleted.

Line 14: "is linear with respect to" -> "scales linearly with the"

We have rewritten: *"hydrogen GWP100 is independent of the size of the emission perturbation"*

Line 15: compositions -> composition (and L.77)

Done.

Line 16: "and it" -> "which" (but whole sentence needs revising)

Rewritten the sentence: *"For methane the CH$_4$ GWP100 increases by up to 3.4 for different future chemical compositions of the atmosphere compared to present-day."*

Line 19-20: Final sentence unclear: reverse ordering and remove negative.

We have reversed the ordering of this sentence.

*"Therefore, when assessing emissions at different locations or for the future with atmospheric composition different from present-day, it is not necessary to adjust the multi-model GWP values."*

Line 23: move "as an energy carrier" to the end of the sentence

Done.

Line 25: composition -> abundance

Done.

Line 36: capture -> captures

Done.

Line 45: "land-ocean fraction" -> "greater landmass"

Done.

Line 46: "higher...in Southern Hemisphere" -> "lower...im Northern Hemisphere"

Done.

Line 54: as -> such as

Done.

Line 55: leads -> lead

Done.

Line 59: remove "as"

Done.

Line 69-70: reverse sentence: sensitivity of GWP to atmospheric composition

Done:

*"Therefore, we investigate the sensitivity of the calculated GWP100 on the chemical composition of the atmosphere using anthropogenic emissions and methane concentrations from three different Shared Socioeconomic Pathways (SSPs)."*

Line 89: anthropogenic -> global anthropogenic (?)

Added "global" to the sentence.

Line 92: emission -> emissions

Done.

Line 98: "first 11" -> "first two sets of" (much clearer to reader!)

Done. Thank you.

Line 102: on -> to

Done.

Line 185: has -> show

Done.

Line 352: "it depends somewhat" -> "are somewhat dependent"

Done.

Line 370: "due to process understanding of" -> "due to weaknesses in understanding of the processes controlling" (or something similar)

As you suggested, we have rewritten the sentence: "*Overall, these dependencies are small compared to the uncertainty in the $H_2$ GWP100 due to weaknesses in understanding of the processes controlling the hydrogen budget.*"

There is an additional reference (Aamaas et al.) at the end of the reference list which is out of place.

The reference management software put Aa at the end of the reference list. Will let the copy-editor decide the location of the reference.

**References:**

Hauglustaine, D., Paulot, F., Collins, W., Derwent, R., Sand, M., and Boucher, O.: Climate benefit of a future hydrogen economy, Communications Earth & Environment, 3,295, 10.1038/s43247-022-00626-z, 2022.

Holmes, C. D., Prather, M. J., Søvde, O. A., and Myhre, G.: Future methane, hydroxyl, and their uncertainties: key climate and emission parameters for future predictions, Atmos. Chem. Phys., 13,285-302, 10.5194/acp-13-285-2013, 2013.

Prather, M. J.: Lifetimes of atmospheric species: Integrating environmental impacts, Geophys. Res. Lett., 29,20-21-20-23, https://doi.org/10.1029/2002GL016299, 2002.

Prather, M. J.: Lifetimes and time scales in atmospheric chemistry, Philosophical Transactions of the Royal Society A: Mathematical, Physical and Engineering Sciences, 365,1705-1726, doi:10.1098/rsta.2007.2040, 2007.

Sand, M., Skeie, R. B., Sandstad, M., Krishnan, S., Myhre, G., Bryant, H., Derwent, R., Hauglustaine, D., Paulot, F., Prather, M., and Stevenson, D.: A multi-model assessment of the Global Warming Potential of hydrogen, Communications Earth & Environment, 4,203, 10.1038/s43247-023-00857-8, 2023.

Szopa, S., V. Naik, B. Adhikary, P. Artaxo, T. Berntsen, W. D. Collins, S. Fuzzi, L. Gallardo, A. Kiendler Scharr, Z. Klimont, H. Liao, N. Unger, and Zanis, P.: Short-Lived Climate Forcers, in: Climate Change 2021: The Physical Science Basis. Contribution of Working Group I to the Sixth Assessment Report of the Intergovernmental Panel on Climate Change, edited by: Masson-Delmotte, V., P. Zhai, A. Pirani, S. L. Connors, C. Péan, S. Berger, N. Caud, Y. Chen, L. Goldfarb, M. I. Gomis, M. Huang, K. Leitzell, E. Lonnoy, J. B. R. Matthews, T. K. Maycock, T. Waterfield, O. Yelekçi, R. Yu, and Zhou, B., Cambridge University Press, Cambridge, United Kingdom and New York, NY, USA 2021.

Thornhill, G. D., Collins, W. J., Kramer, R. J., Olivié, D., O'Connor, F., Abraham, N. L., Bauer, S. E., Deushi, M., Emmons, L., Forster, P., Horowitz, L., Johnson, B., Keeble, J., Lamarque, J. F., Michou, M., Mills, M., Mulcahy, J., Myhre, G., Nabat, P., Naik, V., Oshima, N., Schulz, M., Smith, C., Takemura, T., Tilmes, S., Wu, T., Zeng, G., and Zhang, J.: Effective Radiative forcing from emissions of reactive gases and aerosols – a multimodel comparison, Atmos. Chem. Phys. Discuss., 2020,1-29, 10.5194/acp-2019-1205, 2020.

Warwick, N. J., Archibald, A. T., Griffiths, P. T., Keeble, J., O'Connor, F. M., Pyle, J. A., and Shine, K. P.: Atmospheric composition and climate impacts of a future hydrogen economy, Atmos. Chem. Phys., 23,13451-13467, 10.5194/acp-23-13451-2023, 2023.

---

## Referee Report (RR1)

Comments for Paper entitled "Sensitivity of climate effects of hydrogen to leakage size, location, and chemical background"

**General comments**

Authors have addressed all comment and I think makes the paper much clearer and easier to follow.

I've added a couple of specific comments to clarify a couple of points which aren't immediately clear from reading through.

Overall, I find the manuscript fit for publication and think it will be a valuable asset to the hydrogen community.

**Specific comments**

line 112: For clarity, please specify the impact of NOx:CO to OH e.g. a higher/lower NOx:CO ratio corresponds to a higher/lower OH availability

line 195: Specify these are globally averaged hydrogen concentrations

line 208: I think it's worth adding a line to say that you had calculated the perturbation lifetime with respect to atmosphere and soil separately as this is not immediately clear until after reading the table.

**Technical comments**
lines 317: "Derwent also investigated"

---

## Author Response (AR2)

Once more we would like to thank the two reviewers for their comments on the manuscript. Below follows our responses to the comments by the reviewers and a description of how the manuscript has been modified. The original reviewer's comments are in black and our response in blue.

**Anonymous referee #1**

General comments

Authors have addressed all comment and I think makes the paper much clearer and easier to follow. I've added a couple of specific comments to clarify a couple of points which aren't immediately clear from reading through.

Overall, I find the manuscript fit for publication and think it will be a valuable asset to the hydrogen community.

Specific comments

line 112: For clarity, please specify the impact of NOx:CO to OH e.g. a higher/lower NOx:CO ratio corresponds to a higher/lower OH availability

We have slightly rewritten this sentence, highlighting both the methane concentration and the NOx to CO emission ratio influence on OH, methane lifetime and atmospheric lifetime of OH.

For info, in the introduction there is a similar sentence as the original sentence near line 112: "The $NO_x$ to CO emission ratio has been shown to be important for explaining the OH time evolution and changes in methane lifetime over time (Dalsøren et al., 2016;Skeie et al., 2023)."

The new sentence near line 112 and the specification of the impact of the ratio as well as the methane concentration on OH and methane lifetime.

"The SSPs chosen were based on high and low methane concentration (Fig. 2a) and high and low $NO_x$ to CO emission ratio (Fig. 2b) as both drive changes in OH and methane lifetime and would also influence the atmospheric lifetime of hydrogen. Higher $NO_x$ to CO ratio and lower methane concentration corresponds to increased OH levels and a shorter methane lifetime, while lower ratio and higher methane concentration results in reduced OH levels and a longer methane lifetime."

line 195: Specify these are globally averaged hydrogen concentrations

We have added "globally averaged" to this sentence. "...larger increases in globally averaged hydrogen concentrations..."

line 208: I think it's worth adding a line to say that you had calculated the perturbation lifetime with respect to atmosphere and soil separately as this is not immediately clear until after reading the table.

We have added the following sentence:

"In Table 3, in addition to the total perturbation lifetime, the perturbation lifetime with respect to the atmospheric and soil sink are shown separately."

Technical comments

lines 317: "Derwent also investigated"

Done.

**Anonymous referee #2**

I have reread the revised manuscript and considered the authors responses, and I am satisfied that my concerns have been addressed and that the changes made have improved the manuscript. The upgraded Table 3 adds useful additional information, and the issues with the term "feedback factor" have been successfully resolved. I believe that the paper is now suitable for publication in ACP without substantial further changes, although I have suggested some minor technical corrections and clarifications below.

Minor corrections

The first line of the abstract is still awkward, and it would be better to reverse it: "Use of hydrogen as an energy carrier and reactant in metal production can reduce carbon dioxide emissions by replacing fossil fuel usage."

We have replaced the first sentence in the abstract with your suggestion. Thank you.

Line 15: "For methane the CH4 GWP100" -> "The methane GWP100"

Done.

Line 43: "chemistry-climate model" not needed, as Earth system model is already stated

"chemistry-climate model" deleted.

Line 212: "is lower than" -> "is shorter than". The second half of this sentence would be more clearly phrased as: "…(7 years), it makes a greater contribution to the change in the total lifetime".

The full sentence is now rewritten as:

"As the soil sink lifetime (3.5 years) is shorter than the atmospheric lifetime (7.0 years), it makes a greater contribution to the change in the total perturbation lifetime (Table 3)."

Line 233: "have an extreme location relative to what can be expected" is somewhat unclear, perhaps better as "are remote from locations where hydrogen emissions may be expected to occur".

Thank you for your suggestion. We have rewritten the sentence: "One should note that these two sites are remote from locations where hydrogen emissions may be expected to occur."

Line 254: "productions" -> "production"

Done

Line 313: "on how" -> "to how"

Done.

Line 558: The ACPD discussion paper Thornhill et al. 2020 should be updated to the final ACP paper Thornhill et al., 2021.

The reference is updated.